# IL-37 expression reduces acute and chronic neuroinflammation and rescues cognitive impairment in an Alzheimer's disease mouse model

Niklas Lonnemann[1], Shirin Hosseini[1,2], Melanie Ohm[1,3], Robert Geffers[4], Karsten Hiller[3,5], Charles A Dinarello[6,7]*, Martin Korte[1,2]*

[1]Department of Cellular Neurobiology, Zoological Institute, Braunschweig, Germany; [2]Neuroinflammation and Neurodegeneration Group, Helmholtz Centre for Infection Research, Braunschweig, Germany; [3]BRICS - Braunschweig Integrated Centre of Systems Biology, Braunschweig, Germany; [4]Genome Analytics Group, Helmholtz Center for Infection Research, Braunschweig, Germany; [5]Department of Computational Biology of Infection Research, Helmholtz Centre for Infection Research, Braunschweig, Germany; [6]Department of Medicine, University of Colorado Denver, Aurora, United States; [7]Department of Medicine, Radboud University, Medical Center, Nijmegen, Netherlands

*For correspondence:
cdinare333@aol.com (CAD);
m.korte@tu-bs.de (MK)

**Abstract** The anti-inflammatory cytokine interleukin-37 (IL-37) belongs to the IL-1 family but is not expressed in mice. We used a human IL-37 (hIL-37tg) expressing mouse, which has been subjected to various models of local and systemic inflammation as well as immunological challenges. Previous studies reveal an immunomodulatory role of IL-37, which can be characterized as an important suppressor of innate immunity. Here, we examined the functions of IL-37 in the central nervous system and explored the effects of IL-37 on neuronal architecture and function, microglial phenotype, cytokine production and behavior after inflammatory challenge by intraperitoneal LPS-injection. In wild-type mice, decreased spine density, activated microglial phenotype and impaired long-term potentiation (LTP) were observed after LPS injection, whereas hIL-37tg mice showed no impairment. In addition, we crossed the hIL-37tg mouse with an animal model of Alzheimer's disease (APP/PS1) to investigate the anti-inflammatory properties of IL-37 under chronic neuroinflammatory conditions. Our results show that expression of IL-37 is able to limit inflammation in the brain after acute inflammatory events and prevent loss of cognitive abilities in a mouse model of AD.

## Editor's evaluation

In this manuscript, the authors demonstrated that acute and chronic neuroinflammation was attenuated in human IL-37 (hIL-37) transgenic mice, thus revealing the effects of an anti-inflammatory cytokine hIL-37 in the central nervous system of mice. This study will be of interest to scientists studying neuroinflammation and searching for potential therapeutic targets.

## Introduction

Neuroinflammation is characterized by glial cell activation and is mediated via pro-inflammatory signals (*Ji et al., 2014*; *Rivest, 2009*). In general, acute inflammation is the initial response of the immune

system and is characterized by activation of immune cells, rapid production of various cytokines and chemokines, and phagocytic mechanisms. Although these processes are important to combat pathogens, persistent inflammation can lead to a pathophysiological state that results in tissue damage and loss of function. It is well known that during inflammation, pro- and anti-inflammatory cytokines interact and influence the outcome. Examples of pro-inflammatory cytokines include interleukin (IL)–1β and tumor necrosis factor α (TNF-α), both of which elicit a strong acute inflammatory response (*Dinarello, 1996*; *Leal et al., 2013*). In contrast, anti-inflammatory cytokines such as IL-4, IL-10 (*De Beaux et al., 1996*; *Marie et al., 1996*), or IL-37 (*Nold et al., 2010*) inhibit the action of pro-inflammatory cytokines and hence limit inflammation (*Cavaillon, 2001*). In the CNS, long-term activation of glial cells leads to neuroinflammation, which is an important hallmark of many neurological disorders, including Alzheimer's disease (AD), Parkinson's disease or multiple sclerosis (*Ji et al., 2014*). In particular, AD is the most common form of dementia characterized by amyloid-β plaques and neurofibrillary tangles in the brain tissue. Patients with AD suffer from memory loss, speech disorders, confusion, problems with attention, and spatial orientation. Currently, there are no effective treatment strategies to combat AD despite the considerable clinical need.

Innate immune responses and neuroinflammation are thought to contribute significantly to the progression of AD (*Tan et al., 2007*). Therefore, regulatory cytokines that can reduce inflammation in the CNS are of therapeutic value (*Banchereau et al., 2012*; *Cavalli et al., 2016*). Nevertheless, it remains unclear which of the regulatory mediators act adversely or beneficially at each stage of the neuroinflammatory process (*Ji et al., 2014*; *Grace et al., 2014*; *Kigerl et al., 2009*).

For example, the release of IL-1β under pathological conditions can lead to deficits in learning and memory processes as well as in long-term potentiation (LTP) (*Avital et al., 2003*; *Hein et al., 2010*; *Tong et al., 2012*). In addition, pro-inflammatory cytokines such as TNF-α and interferon-γ (IFN-γ) have been associated with impairments in hippocampal neuron structure and function during viral infection (*Hosseini et al., 2018*). It is important to note that in chronic inflammation, the homeostasis of pro- and anti-inflammatory mediators is completely disturbed and must be therapeutically directed at attenuating persistent inflammatory processes. Among the important pro-inflammatory mediators, both IL-1α and IL-1β play a role in autoinflammatory, autoimmune, infectious, and degenerative diseases (*Dinarello, 2009*; *Dinarello, 2010*; *Dinarello et al., 2012*; *Gabay et al., 2010*; *Sims and Smith, 2010*). Although IL-1α and IL-1β are encoded by different genes, both cytokines are structurally related proteins that bind to the type I interleukin 1 receptor (IL-1R1) and elicit similar innate responses. However, IL-1α is constitutively present in healthy cells but is only released under cell stress conditions, as is the case during inflammation. Moreover, because it is constitutively present, IL-1α acts rapidly to trigger local inflammation. In contrast, IL-1β is not present in the healthy state; the IL-1β precursor is not active but requires processing by caspase-1, an intracellular protease, resulting in conversion to an active cytokine that is secreted (*Garlanda et al., 2013*).

Unlike most members of the IL-1 family that induce inflammation, this important cytokine family also includes two members (IL-37 and IL-38) that act as anti-inflammatory signalling agents and are thought to dampen an ongoing innate immune response (*Nold et al., 2010*; *Dinarello, 2018*). In particularly, IL-37 serves as a potent mediator to limit the inflammatory responses (*Cavalli and Dinarello, 2018*; *Caraffa et al., 2018*). Therefore, a transgenic mouse model expressing the human splice variant IL-37b (hIL-37tg) was developed (*Nold et al., 2010*; *Cavalli and Dinarello, 2018*) and showed potent anti-inflammatory and beneficial effects in a wide variety of different pathological conditions in different organs (*Nold et al., 2010*; *Ballak et al., 2014*; *Coll-Miró et al., 2016*; *McNamee et al., 2011*; *Yousif et al., 2011*). The transgenic mouse was generated to allow continuous expression of IL-37 in almost all cells. This was achieved by using the full-length cDNA of IL-37b through the CMV promoter (*Nold et al., 2010*). It is important to note that these IL-37tg mice did not exhibit an abnormal phenotype and reproduced normally (*Nold et al., 2010*; *Dinarello et al., 2016*). IL-37 has been described as a protein that binds to the IL-18 receptor (IL-18R) and, unlike IL-18, does not elicit a pro-inflammatory immune response but prevents it and is even involved in an anti-inflammatory response (*Cavalli and Dinarello, 2018*). IL-37 binds the alpha chain of the IL-18R and additionally interacts with IL-1R8 (also known as TIR8 or SIGIRR) rather than recruiting the beta chain of the IL-18R (as is the case with IL-18 binding) (*Nold-Petry et al., 2015*). This co-localization was also demonstrated in mouse cells (*Nold-Petry et al., 2015*). Other studies showed the mRNA of IL-18Rα and SIGIRR receptors on astrocytes and microglia (*Andre et al., 2005*). Notably, IL-18Rα has been shown to be expressed on astrocytes

and microglia, and IL-18 is produced only by microglial cells, suggesting a signalling cascade running mainly through microglia with respect to IL-18 and IL-37 (*Andre et al., 2005*; *Tsilioni et al., 2019*). In addition, two recent studies focusing on the central and peripheral nervous systems have demonstrated the therapeutic potential of IL-37 in autism spectrum disorders (ASD) and multiple sclerosis (MS) (*Tsilioni et al., 2019*; *Cavalli et al., 2019*). *Tsilioni et al., 2019* have demonstrated a reducing effect of inflammation on IL-37 expression in cultured human microglia.

An important factor for the expression of IL-37 related to the instability sequence was also discovered in a mouse cell line. *Bufler et al., 2004* showed that despite strong activation of the CMV promoter, expression of IL-37 could not be detected (*Bufler et al., 2004*). It was also shown that the specific mRNA was rapidly degraded. However, a significant and rapid increase in IL-37 mRNA could be achieved by an LPS stimulus (*Bufler et al., 2004*). Similarly, administration of recombinant IL-37 frequently exhibited beneficial effects on the outcome of various animal models of human diseases (*Cavalli et al., 2016*; *Coll-Miró et al., 2016*; *Moretti et al., 2014*; *Wu et al., 2014*). In addition, IL-37 has been shown to regulate cellular metabolism after an inflammatory stimulus (*Su and Tao, 2021*). In the last decade, metabolic changes have been described as an important determinant of immunological processes, as shown in macrophages after LPS stimulation (*Su and Tao, 2021*; *O'Neill and Hardie, 2013*). The LPS-induced increase in mTOR phosphorylation and decreased AMP-activated protein kinase activity were reversed by expression of IL-37 (*Nold et al., 2010*; *Ballak et al., 2014*; *Nold-Petry et al., 2015*; *Su and Tao, 2021*).

In the present study, we used hIL-37tg mice to investigate the effect of IL-37 expression on the acute inflammatory processes induced by LPS or IL-1β administration. In addition, we crossed the APP/PS1 mouse model of AD with hIL-37tg mice to investigate the role of IL-37 expression during the chronic state of neuroinflammation and the consequences for the progression of AD pathology.

## Results

### Primary microglia from IL-37tg mice exhibit an inflammatory-suppressive response after LPS challenge in vitro

To investigate the effect of IL-37 on microglial cell activation, the release of the pro-inflammatory cytokines IL-6, IL-1β, and TNF-α was analyzed after LPS stimulation of wild-type (WT) and IL-37 transgenic microglial cells (*Figure 1A*).

First, IL-6 secretion in heterozygous (IL-37^wt/tg) and homozygous (IL-37^tg/tg) IL-37 transgenic primary microglia was analyzed and compared to IL-6 secretion of WT microglia upon LPS stimulation. The results showed that the amounts of IL-6 secreted by microglial cells from homozygous IL-37 transgenic (IL-37^tg/tg) mice were significantly reduced compared to microglial cells from control mice after 24 hr of stimulation with 100 ng/ml and 1 µg/ml LPS and additionally also 6 hr after stimulation with 1 µg/ml LPS ($F_{(2,11)}=7.941$ p=0.007; p=0.007; p=0.054; $F_{(2,11)}=3.915$ p=0.052; p=0.043; p=0.359; $F_{(2,9)}=12.74$ p=0.002; p=0.004; p=0.009) (*Figure 1B*). Microglial cells from heterozygous IL-37 transgenic mice (IL-37^wt/tg) also showed decreased IL-6 secretion, but only after 6 hr of stimulation with 1 µg/ml LPS ($F_{(2,9)}=12.74$ p=0.002; p=0.009) (*Figure 1B*). Thus, the immunosuppressive effect of transgenic expression of IL-37 on primary microglia was demonstrated only in this experiment by differences between hetero- and homozygous IL-37 transgenic mice. All subsequent experiments were analyzed using homozygous (IL-37^tg/tg) IL-37 transgenic mice (referred to as IL-37tg). Further analysis also confirmed the reduction in inflammatory mediators by measuring the release of IL-6 when other concentrations of the inflammatory stimulus were examined (*Figure 1C*) and could be extended by the significantly reduced release of TNF-α (*Figure 1D*) and IL-1β (*Figure 1E*), compared with primary microglial cells from control mice (IL-6: p=0.1007; p=0.0024; p=0.0001; TNF-α: p=0.0038; p<0.0001; p<0.0001; IL-1β: p=0.0295; p=0.0001; p<0.0001). Remarkably, LPS stimulation caused a detectable increase in the level of transgenic *IL37* mRNA after 1 hr, which remained elevated in the following hours up to 24 hr post stimulation ($F_{(7,15)}=2.629$ p=0.05509; p=0.0171, p=0.0086, p=0.0238) (*Figure 1F*). As mentioned previously, the amount of IL-6 protein in the supernatant of the same cultures was significantly reduced when IL-37tg microglial cells were compared to identically stimulated control cells (p<0.0001) (*Figure 1G*). Examining inflammatory responses in primary astrocytes as a different group of glial cells, we did not detect decreased levels of IL-6 and TNF-α in IL-37tg cells 24 hr after

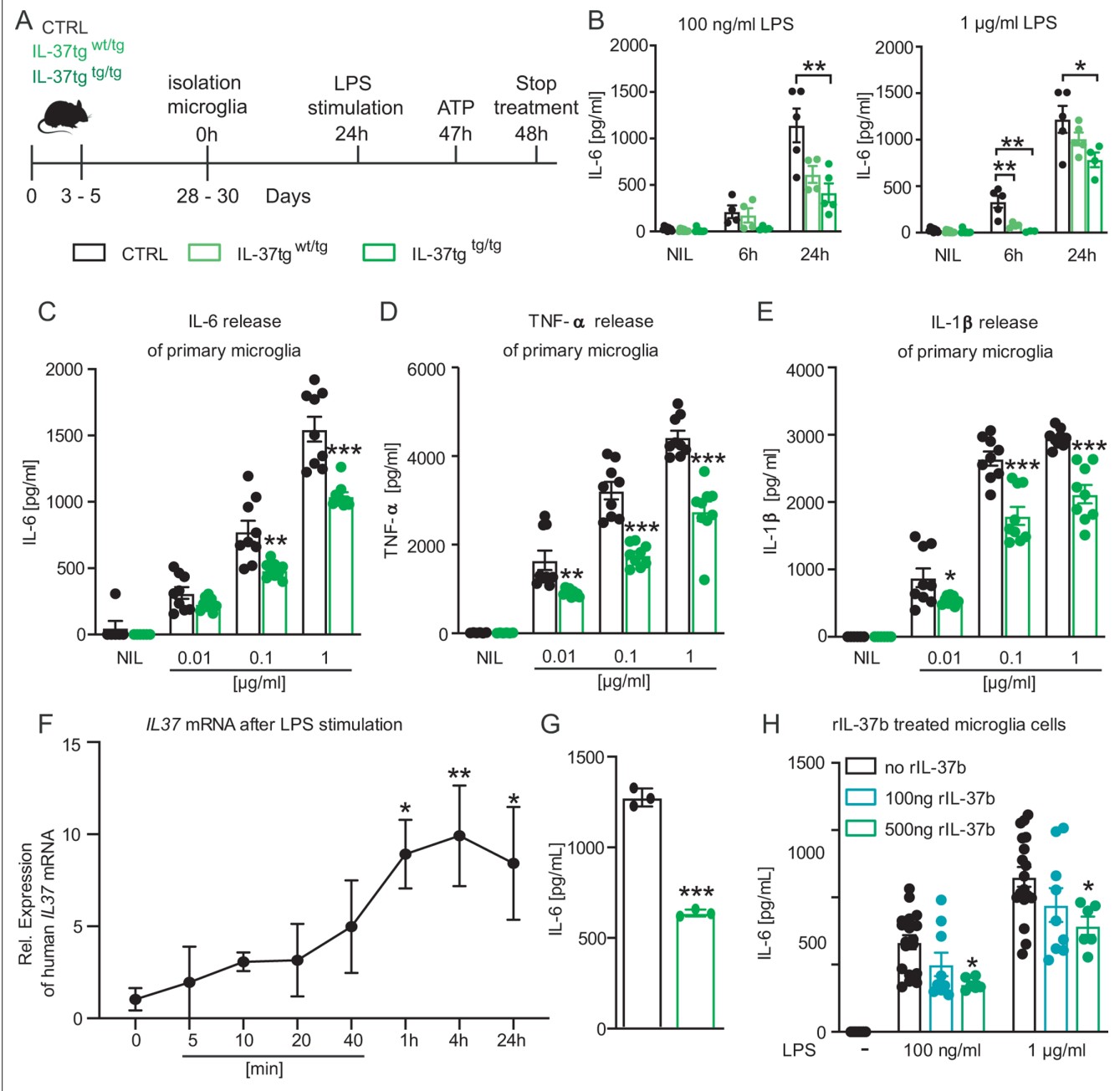

**Figure 1.** Primary microglial cells from IL-37tg mice showed decreased pro-inflammatory cytokine release after stimulation by LPS. (**A**) WT and IL-37tg primary microglial cells (P3-5) were plated and stimulated without LPS or with LPS. Cells were treated for 6 hr (**B**) and 24 hr (**B–E**). Cells from homozygous transgenic animals released less pro-inflammatory cytokines IL-6 (**C**), TNF- (**D**) and IL-1β (**E**) under different LPS concentration (B-E n=4–9). (**F**) IL37 mRNA was analyzed in IL-37tg microglial cells at specific intervals after addition of LPS. (**G**) Levels of IL-6 were measured from the same cells as in (**F**) and were detectable only after 24 hr (n=3). (**H**) Levels of IL-6 were measured in WT after addition of increasing concentrations of recombinant IL-37b and 100 ng LPS (H n=6–18). Data are presented as mean ± SEM. * p<0.05, ** p<0.01, *** p<0.001 compared to WT. (B+F + H: 1-way ANOVA with multiple comparison; C-E+G: t-test).

The online version of this article includes the following source data and figure supplement(s) for figure 1:

**Source data 1.** Release of pro-inflammatory cytokines by microglia after LPS stimulation.

**Figure supplement 1.** Pro-inflammatory cytokine release by primary astrocytes after LPS stimulation.

**Figure supplement 1—source data 1.** Release of pro-inflammatory cytokines by astrocytes after LPS stimulation.

LPS stimulation (100 ng/ml) compared with WT primary astrocytic cells (*Figure 1—figure supplement 1A,B*), confirming the importance of anti-inflammatory IL-37 signaling, which may proceed via microglia but not astrocytes.

To investigate the potential effect of recombinant IL-37 (rIL-37) in inhibiting LPS-induced release of proinflammatory cytokines (here IL-6) by microglia, primary WT microglial cells were pretreated with either 100 ng/mL or 500 ng/mL rIL-37 for 2 hr and then stimulated with increasing concentrations of LPS. Pretreatment with rIL-37 was chosen because microglial cells from IL-37tg mice had very low basal expression of IL-37 even in the absence of LPS (*Figure 1F*). Therefore, the response of WT microglial cells to LPS in the presence of different concentrations of rIL-37 was examined here. After LPS induction, microglial cells pretreated with the higher concentration of rIL-37 showed a significant decrease in the release of IL-6 compared with cells pretreated with rIL-37 vehicle (PBS) ($F_{(2,30)}=5.135$ p=0.0121; p=0.0133; $F_{(2,30)}=3.512$ p=0.0426; p=0.0498) (*Figure 1H*).

Taken together, these results indicate that microglial cells in particular release lower levels of pro-inflammatory cytokines upon acute inflammatory stimulation in the presence of IL-37.

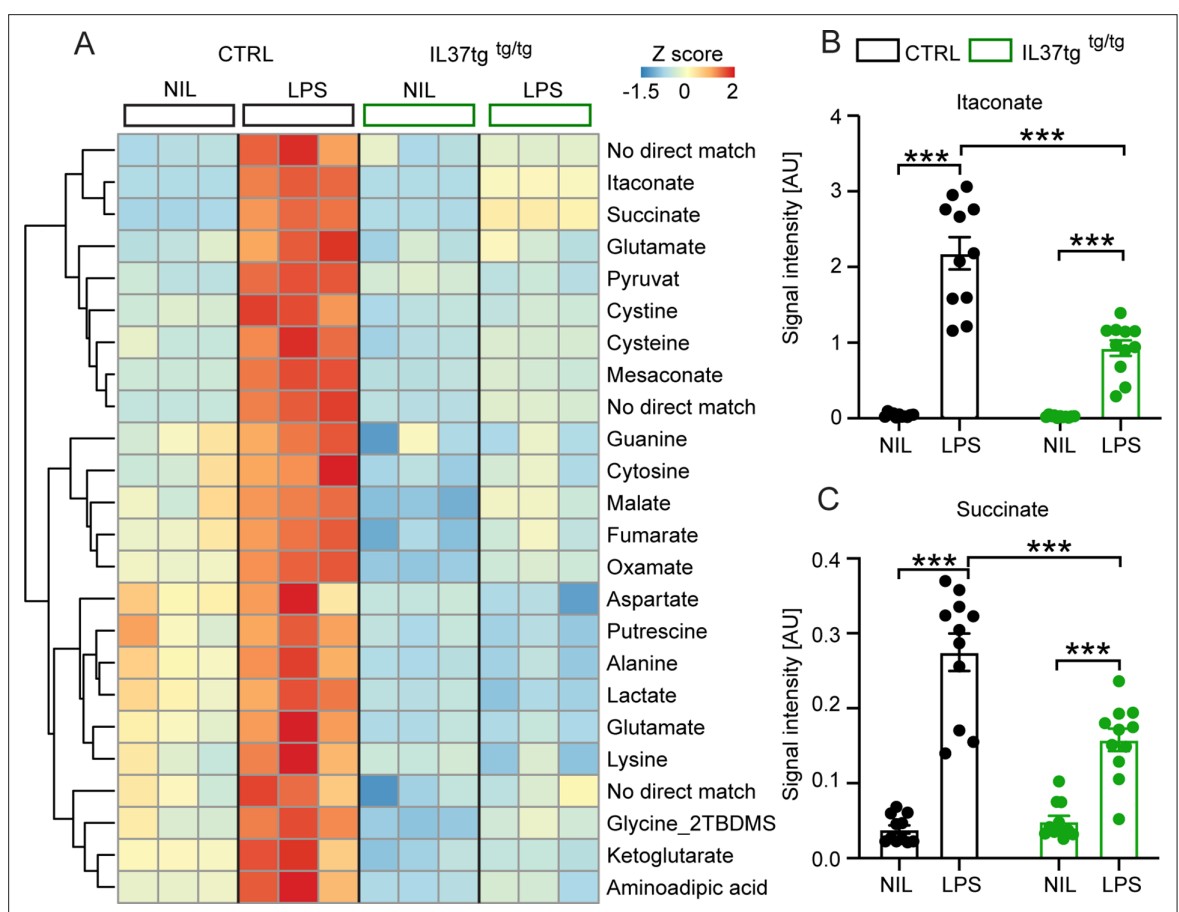

**Figure 2.** Primary microglial cells isolated from IL-37tg mice exhibited reduced levels of inflammation-associated intracellular metabolites after stimulation with LPS. WT and IL-37tg microglia were stimulated with 10 ng/ml LPS (based on highly sensitive metabolomics assessments). Metabolomic analysis was performed on these cells. (**A**) Heatmap of identified significantly altered metabolites after or without LPS stimulation (described as Z-score). (**B–C**) Significant effects in WT, treated with LPS, were seen with respect to itaconate (**B**) and succinate (**C**), whereas these changes were significantly reduced in IL-37tg microglial cells treated with LPS compared with WT cells (n=11). Data are presented as mean ± SEM. *** p<0.001 (two-way ANOVA with multiple comparison).

The online version of this article includes the following source data for figure 2:

**Source data 1.** Inflammation-associated intracellular metabolites in primary microglia after LPS stimulation.

## Metabolomic profiling of microglial cells from IL-37tg mice reveals an attenuated metabolic response associated with inflammation after LPS stimulation in vitro

Recent studies have discovered several metabolic intermediates that contribute directly to immune function. Among them, the TCA cycle-related metabolites succinate and itaconate are novel markers associated with pro-inflammatory macrophage activation and function (*He et al., 2021*; *Michelucci et al., 2021*; *Mills et al., 2021*). To address the question of whether microglial cells expressing the anti-inflammatory cytokine IL-37 are further capable of modulating inflammation-related metabolic changes, we performed intracellular metabolomic analysis of control and IL-37tg microglial cells treated with 10 ng/ml LPS for 24 hr. As expected, many metabolites were significantly elevated in LPS-treated control cells, including itaconate and succinate, both metabolic markers for proinflammatory activation of macrophages (*Figure 2A*; *Michelucci et al., 2021*; *Tannahill et al., 2013*). Levels of both metabolites were significantly less elevated in IL-37tg microglial cells, indicating an attenuated pro-inflammatory response at the metabolic level (*Figure 2B and C*) (itaconate $F(1,20)=174.9$ $p<0.0001$; $p<0.0001$; $p<0.0001$; $F(1,20)=27.55$ $p<0.0001$; $p<0.0001$; succinate $F(1,20)=98.40$ $p<0.0001$; $p<0.0001$; $p=0.0005$; $F(1,20)=17.01$ $p=0.0005$; $p<0.0001$). This is consistent with the profiles of other metabolites of central carbon metabolism showing modest effects after LPS treatment (*Figure 2A*). We conclude that IL-37tg microglial cells release lower levels of pro-inflammatory cytokines than control cells after LPS stimulation and that IL-37 expression in microglial cells limits LPS-induced pro-inflammatory metabolic reprogramming.

## Microglial activation and inflammatory responses are reduced after LPS challenge in IL-37tg mice

To test whether the anti-inflammatory properties of IL-37 on microglial cells observed in vitro studies, are similar in vivo, we intraperitoneally injected adult (3–8 month-old) homozygous IL-37tg mice (referred to as IL-37tg) and age-matched control animals with either saline (vehicle control group) or LPS ($2\times0.5$ mg/kg) (*Figure 3A*). Both experimental groups showed significant weight loss in response to systemic LPS injection, but the effect was significantly higher in control mice than in IL-37tg mice (weight loss $F(1,17)=220.5$ $p<0.0001$; $p<0.0001$; $p<0.0001$; $F(1,41)=1.486$ $p=0.2299$; $p=0.0139$) (*Figure 3B*). We next examined the effect of systemic administration of LPS on neuroinflammation by using brain homogenates for ELISA assays and also by isolating microglia from the brains of controls and LPS-treated animals for FACS analyzes. Control mice treated with LPS had a significantly higher percentage of CD68-expressing microglial cells (gated on the CD11b[+]/CD45[low] population) (CD68 $F(1,6)=19.52$ $p=0.0045$; $p=0.0017$; $p>0.9999$; $F(1,6)=48.42$ $p=0.0004$; $p=0.0001$) (*Figure 3C–D*) compared to the cells of control animals treated with saline. Similarly, IL-1β levels in the brain of LPS-treated WT mice were significantly higher than those of saline-treated WT controls (IL-1β $F(1,7)=9.399$ $p=0.0182$; $p=0.0244$; $p=0.7541$; $F(1,7)=3.182$ $p=0.0935$; $p=0.0366$) (*Figure 3E*). Remarkably, IL-37tg animals showed no significant changes in CD68-expressing microglia and IL-1β levels when challenged with LPS in the same manner compared with matched controls (*Figure 3C–E*). Further analysis of pro-inflammatory cytokines IL-6 and TNF-α showed no significant changes between WT and IL-37tg mice treated with LPS and control groups, but a trend toward higher levels of IL-6 was observed in WT brain homogenates after LPS challenge (*Figure 3—figure supplement 1A,B*). Further investigation of cell populations in the brains of WT and IL-37tg mice after LPS stimulation underscored the importance of the IL-37 response, particularly in microglial cells (*Figure 3—figure supplement 2A-E*), whereas macrophages (gated on the CD11b[+]/CD45[high] population) showed no changes in CD68-expressing cells after LPS stimulation or between genotypes (*Figure 3—figure supplement 2F-H*). In addition, analysis of cell numbers in all experimental groups revealed a generally high number of microglial cells in the brain of the animals (*Figure 3—figure supplement 2D and K*) and, in contrast, very low numbers of macrophages (*Figure 3—figure supplement 2G and K*) and leukocytes (gated on the CD11b[-]/CD45[high] population) (*Figure 3—figure supplement 2J-K*), again highlighting the important role of microglia in this scenario.

The activation feature of microglial cells is reflected in the total number of microglial cells in the brain parenchyma and the number of primary processes of these CNS resident immune cells. For example, a higher number of IBA-1-positive cells and a reduced number of primary processes correlate with increased microglial activation (*Hanisch and Kettenmann, 2021*; *Papageorgiou et al.,*

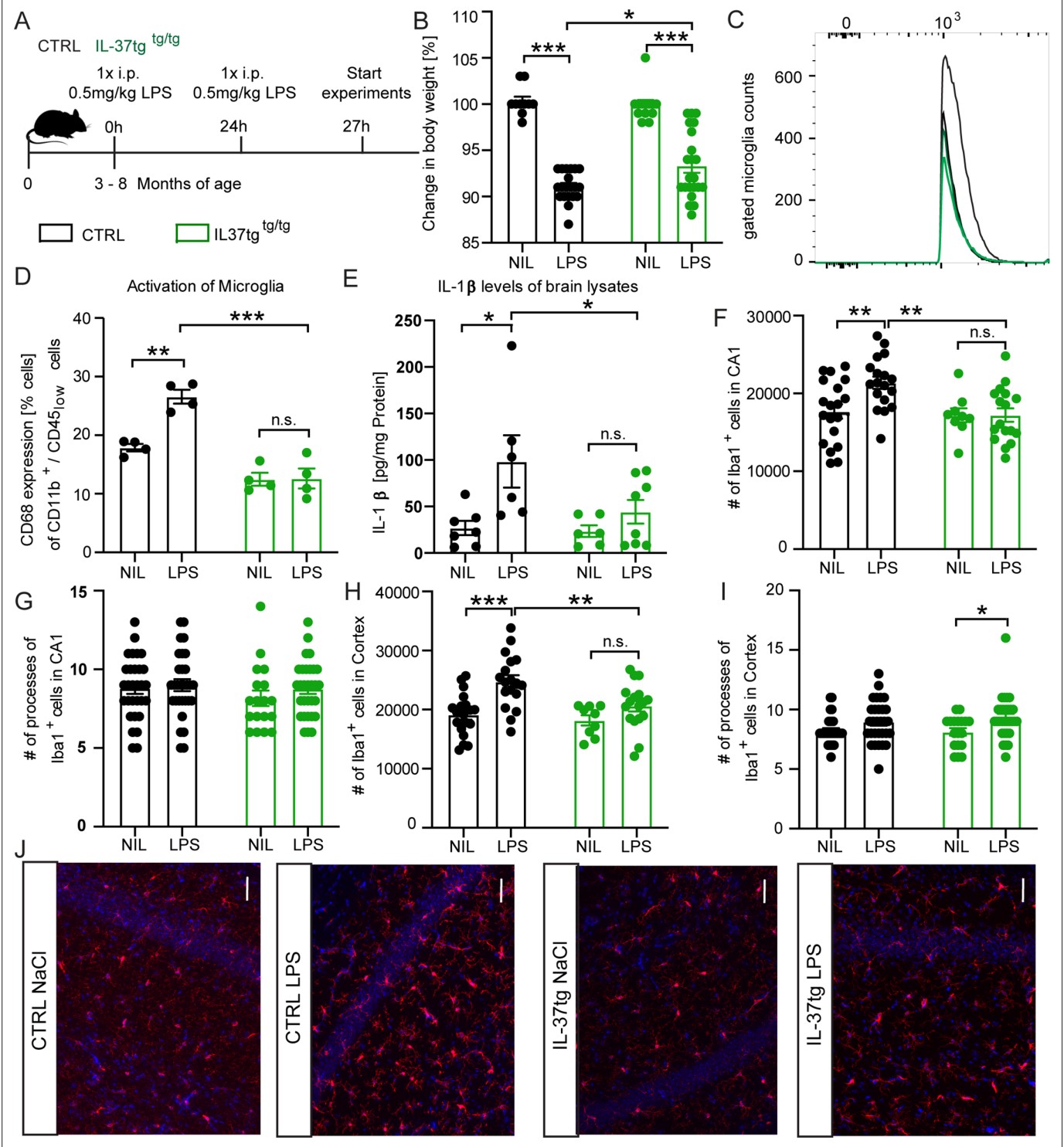

**Figure 3.** IL-37tg mice showed decreased pro-inflammatory cytokine release and less activated microglia after the stimulus of LPS. (**A**) WT and IL-37tg mice were stimulated with saline or LPS. (**B**) IL-37tg animals exhibited significantly less weight loss compared to WT mice. However, IL-37tg mice also had a significant weight change compared to saline treated mice (n=10–22). Microglial cell activation was analyzed by FACS method. (**C–D**) Microglial cells were identified as CD11b⁺ and CD45^low cells and analyzed for CD68 expression. IL-37tg mice had a lower percentage of cells with CD68 expression compared with WT mice after LPS stimulation (n=4) (**C–D**). (**E**) In addition, WT mice exhibited a significant increase in IL-1β levels after LPS treatment, whereas IL-37tg mice did not (n=6–8). (**F–I**) Morphological analysis of microglial cells showed an increased number of IBA-1-positive cells in WT animals treated with LPS compared with saline-treated animals. In contrast, there is no increased IBA-1-positive cell number in IL-37tg animals after

*Figure 3 continued on next page*

*Figure 3 continued*

LPS stimulation (n=9–18; n=18–30 cells for processes). (**J**) Representative images of IBA-1-positive cells (red) and DAPI (blue); scale bar 40 μm. Data are shown as mean ± SEM. * p<0.05, ** p<0.01, *** p<0.001, (B-I: 2-way ANOVA with multiple comparison).

The online version of this article includes the following source data and figure supplement(s) for figure 3:

**Source data 1.** Neuroinflammatory status of mouse brain after systemic LPS challenge.

**Figure supplement 1.** Pro-inflammatory cytokines in brain lysates after peripheral LPS challenge.

**Figure supplement 1—source data 1.** Pro-inflammatory cytokines in mouse brain after systemic LPS challenge.

**Figure supplement 2.** FACS analysis of brain cells after peripheral LPS administration.

---

*2015*; *Wolf et al., 2017*). Therefore, we performed immunostaining with the known microglial marker IBA-1 on brain sections obtained from the animals of both genotypes treated with either saline or LPS (*Figure 3F–J*). A significant increase in the number of IBA-1 positive cells was observed in the CA1 subregion of the hippocampus and in the cortex of LPS-treated animals compared with saline-treated control animals. However, in the IL-37tg mice, LPS did not result in a significant increase in the number of microglial cells (IBA-1 CA1 $F_{(1,25)}=5.222$ p=0.0311; p=0.0024; p>0.9999; $F_{(1,34)}=4.951$ p=0.0328; p=0.0027;; IBA-1 Cx $F_{(1,25)}=15.97$ p=0.0005; p=0.0002; p=0.3234; $F_{(1,34)}=5.159$ p=0.0296; p=0.0054) (*Figure 3F and H*). Further analysis of the number of microglial primary processes showed no differences between animals in the control and the IL-37tg groups treated with either saline or LPS, except that microglial primary processes were significantly increased in the cortex of IL-37tg mice treated with LPS compared with saline-treated IL-37tg mice, which may indicate more branched microglial features (processes CA1 $F_{(1,46)}=1.066$; p>0.9999; p=0.6229; processes Cx $F_{(1,104)}=9.865$; p=0.1401; p=0.0232) (*Figure 3G1*). Overall, these results demonstrate the anti-inflammatory effect of IL-37 expression on brain after LPS challenge, possibly mediated via microglial cells.

## IL-37tg mice are protected from functional and structural neuronal deficits after LPS challenge

Neuroinflammation has been shown to impair hippocampal network function (*Hosseini et al., 2018*; *Beyer et al., 2020*). Because previous studies indicate a significant induction of neuroinflammation after systemic LPS challenge, we hypothesized that IL-37 might also have a beneficial effect on hippocampal network function and structure. Therefore, as described above, we injected control and IL-37tg animals with either saline or LPS for subsequent analysis of neuronal function and structure. We, first examined long-term synaptic plasticity, the ability of synapses to change their transmission strength, which is considered a cellular correlate of learning and memory processes (*Bliss and Collingridge, 1993*). To this end, we induced long-term potentiation (LTP) at the Schaffer collateral CA3 to CA1 pathway in the hippocampus. After 20 minutes of baseline recording, we observed significantly impaired LTP in the acute hippocampal slices of control mice treated with LPS compared to saline ($F_{(1,28)}=4.459$ p=0.0438) (*Figure 4A*). In contrast, IL-37tg animals treated with LPS did not show comparable deficits in synaptic plasticity compared to saline treated IL-37tg animals ($F_{(1,23)}=0.0849$ p=0.7734) (*Figure 4B*). These differences were also evident in the maintenance phase of LTP (last 5 min of the measurement) (last 5 min $F_{(1,22)}=7.887$ p=0.0102; p=0.0043; p>0.9999) (*Figure 4C*).

To assess the effects of IL-37 on neuronal structure, we analyzed the dendritic spine density of hippocampal neurons from all experimental groups. Dendritic spines are small protrusions representing the postsynaptic part of excitatory synapses and were counted on the apical dendrites of CA1 neurons as well as on the dendrites of dentate gyrus neurons. A significant decrease in the density of dendritic spines of CA1 pyramidal neurons and dentate gyrus cells was observed in control mice treated with LPS compared with saline-treated mice (spines CA1 $F_{(1,32)}=22.81$ p<0.0001; p<0.0001; p=0.1145; $F_{(1,44)}=26.68$ p<0.0001; p<0.0001; spines DG $F_{(1,26)}=3.307$ p=0.0805; p=0.0062; p>0.9999; $F_{(1,36)}=0.529$ p=0.4718; p=0.0295) (*Figure 4D–F*).

Similar to the LTP data, the density of dendritic spines of neurons from IL-37tg animals was unchanged when LPS- and saline-injected mice were evaluated (*Figure 4D–F*). In conclusion, these results demonstrate a negative effect of systemically administered LPS on synaptic plasticity that can be restored by IL-37.

Given the possibility that peripheral immune cells in IL-37tg mice also produce IL-37 after LPS challenge and already attenuate the systemic immune response, the question arose whether IL-37 would

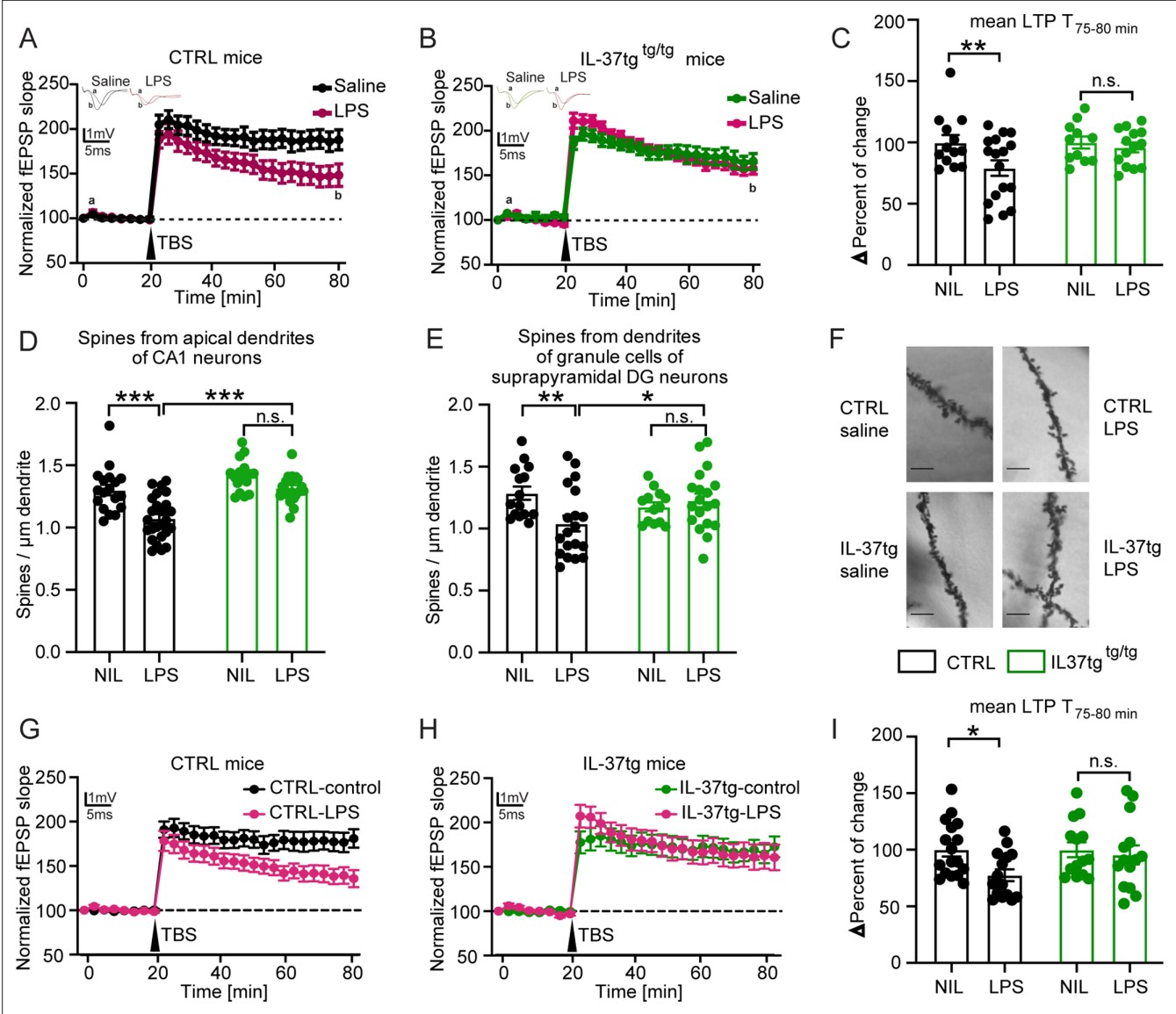

**Figure 4.** IL-37tg mice showed rescued synaptic plasticity and restored loss of spine density after stimulation by LPS compared with WT animals. (**A**) WT animals stimulated with LPS showed significant impairment of theta burst stimulation-induced LTP (TBS) compared with WT, which were treated with saline. (**B**) In contrast, IL-37tg mice showed no significant impairment of LTP after LPS treatment. (**C**) Mean LTP magnitude (average of 55–60 min after TBS) was significantly lower in WT mice treated with LPS, while IL-37tg mice showed no significant differences (n of mice 3–4; n of acute slices 11–17). (**D–F**) Spine density in apical dendrites of the CA1 hippocampal neurons and in the superior DG neurons was significantly decreased in WT mice treated with LPS, whereas spine density of IL-37tg animals treated with LPS was not affected (n of mice 3–4; n of dendrites 13–25) (**D and E**). (**F**) Representative images of dendritic spines of hippocampal CA1 neurons in the tested groups were shown; scale bar 5 μm. (**G**) WT acute slices stimulated with LPS showed significant impairment of TBS-induced LTP compared with WT acute slices treated with ACSF. (**H**) In contrast, acute slices from IL-37tg mice showed no significant impairment of LTP after LPS treatment. (**I**) Mean LTP magnitude (mean of 55–60 min after TBS) was significantly lower in acute slices from WT mice treated with LPS, whereas slices from IL-37tg mice showed no significant differences (n of mice 3–4; n of acute slices 14–17). Data are presented as mean ± SEM. * p<0.05, *** p<0.001, (A-I: two-way ANOVA with multiple comparison).

The online version of this article includes the following source data for figure 4:

**Source data 1.** Assessment of synaptic plasticity after LPS challenge.

have a similar protective effect directly in the CNS after LPS stimulation. To accurately demonstrate the beneficial effect of IL-37 in the CNS without the influence of peripheral cells, another electrophysiological experiment was performed.

Here, acute hippocampal slices from control and IL-37tg animals were prepared and stimulated with LPS (10 μg/ml) in ACSF for 2 hr after a resting period before recording. In the control groups,

the sections were kept in ACSF only. These further results indicated that direct IL-37 expression in the acute slices of IL-37tg mice appears to be sufficient to reverse the impairments in LTP after LPS administration. This is because, in contrast to the significant impairment of LTP after LPS administration in acute hippocampal slices of control mice (LTP WT F(1,30)=4.925 p 0.05) (*Figure 4G*), no impairment of LTP was observed in acute slices of IL-37tg mice when treated with LPS (LTP IL-37tg F(1,26)=0.0139 p=0.9069) (*Figure 4H*). This effect was also evident in the data for the last 5 min of the measurement, which represents the maintenance phase of LTP (last 5 min F(1,27)=4.613 p=0.0409; p=0.0285; p 0.9999) (*Figure 4I*). These results clearly demonstrate that local IL-37 expression in the brain can prevent the deleterious effects of LPS on neuronal function.

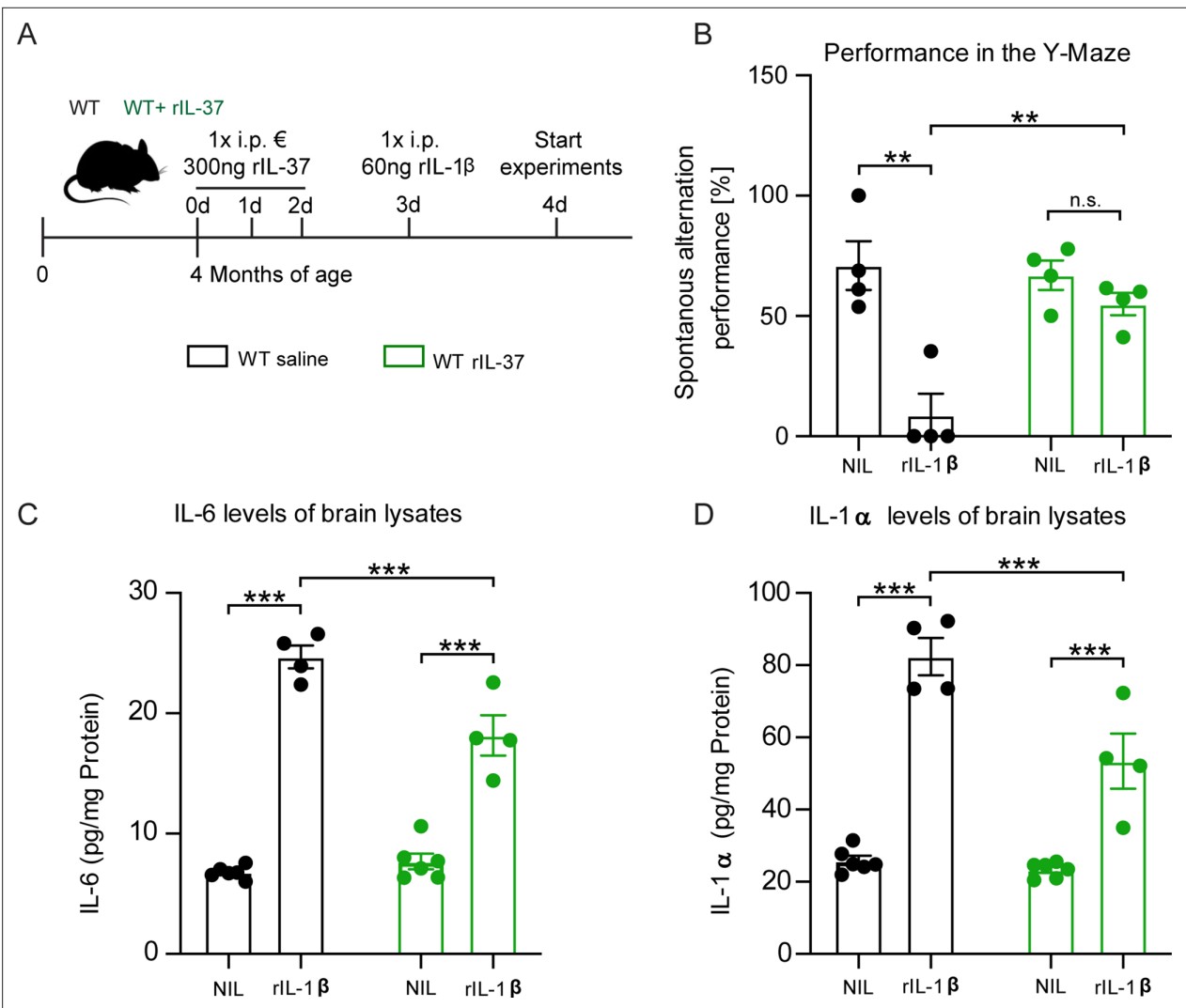

**Figure 5.** Injection of recombinant IL-37 into WT mice showed restoration of cognitive deficits and reduced release of pro-inflammatory cytokines after IL-1β-mediated immunostimulation. (**A**) WT mice were pretreated with either saline or rIL-37 for three consecutive days and then injected with saline or IL-1β. (**B**) WT mice pretreated with saline and then stimulated with IL-1β failed to perform the Y-maze test, whereas WT mice pretreated with rIL-37 and then stimulated with IL-1β performed the test without deficits (n=4). (**C and D**) Although the pro-inflammatory cytokine levels of IL-6 and IL-1α were significantly increased in stimulated WT mice pretreated with rIL-37 compared with the control group, the mice treated with rIL-37 showed a significant decrease in cytokine levels after immunostimulation with IL-1β compared with saline treated group (n=4–6). Data are presented as mean ± SEM. ** p<0.01, *** p<0.001, (B-D: two-way ANOVA with multiple comparison).

The online version of this article includes the following source data and figure supplement(s) for figure 5:

**Source data 1.** Suppressive effect of rIL-37 on the inflammatory response induced by IL-1β.

**Figure supplement 1.** Pro-inflammatory cytokines in brain lysates after peripheral pretreatment with recombinant IL-37 followed by LPS challenge.

**Figure supplement 1—source data 1.** Pro-inflammatory cytokine levels induced by LPS in the brain of rIL-37 treated mice.

## Recombinant IL-37 reduces inflammatory response in vivo and alleviates short-term memory impairment induced by pro-inflammatory cytokine stimulation

To investigate the anti-inflammatory properties of the recombinant IL-37 (rIL-37) protein in vivo, wild-type mice were pretreated with either 300 ng rIL-37 per animal (i.p.) or an equivalent amount of vehicle (saline, control group) for 3 consecutive days. On day 4, animals were injected with 60 ng i.p. IL-1β or saline as control (*Figure 5A*). After another 24 hr, the animals were trained and tested with the Y-Maze behavioral test to assess short-term memory based on the mice's natural willingness to explore a new area. The score for spontaneous alternation depends on the mouse's tendency to seek out a less recently entered arm of the maze. Therefore, this test also measures spatial hippocampus-dependent memory function.

In the absence of rIL-37 pretreatment, we detected a significant performance deterioration in IL-1β-injected mice compared to the corresponding saline-injected control group (Y-Maze $F_{(1,6)}=18.25$ p=0.0052; p=0.0046; p=0.7313) (*Figure 5B*). In contrast, pretreatment with rIL-37 protected the mice from the behavioral deficits induced by IL-1β administration ($F_{(1,6)}=9.899$ p=0.0199; p=0.0024). To analyze the inflammatory mediators in the CNS of these animals, the levels of the pro-inflammatory cytokines IL-6 and IL-1α were measured in brain lysates. We observed that animals stimulated with rIL-1β showed significantly elevated levels of both cytokines compared with the corresponding controls (IL-6 $F_{(1,16)}=264.6$ p<0.0001; p<0.0001; p<0.0001; $F_{(1,16)}=10.27$ p=0.0055; p=0.0004; IL-1α $F_{(1,16)}=132.0$ p<0.0001; p<0.0001; p<0.0001) (*Figure 5C–D*). However, pretreatment with rIL-37 resulted in significantly lower levels of IL-6 and IL-1α in the brains of rIL-1β immunostimulated mice ($F_{(1,16)}=17.47$ p=0.0007; p=0.0003) (*Figure 5C–D*). Taken together, these data indicate the beneficial effects of IL-37 expression on cognitive function in immunostimulated mice.

To investigate the possible preventive effect of rIL-37 on LPS-induced neuroinflammation, in addition to IL-1β-immunostimulation, WT mice were pretreated with either 100 ng rIL-37 per animal (i.p.) or an equivalent amount of vehicle (saline, control group) for 3 consecutive days. On day 3 and 4, the animals were injected twice with LPS (0.5 mg/kg) and 3 hr after the last injection, the level of proinflammatory cytokines in the brain was measured (*Figure 5—figure supplement 1A-C*). The results showed that although injection of LPS in the saline-treated mice resulted in a significant increase in the levels of IL-1β (p<0.01), IL-6 (p=0.04), and TNF-α (p<0.004), only the production of IL-1β was significantly increased in the mice receiving rIL-37 (p=0.001), and the levels of TNF-α (p=0.97) and IL-6 (p=0.63) were not significantly increased in the brains of the mice pretreated with rIL-37 (*Figure 5—figure supplement 1A-C*). These results also highlight the protective role of rIL-37 in modulating LPS-induced proinflammatory cytokines.

## IL-37 shows beneficial effects on neuronal deficits and microglia activation in APP/PS1-IL37tg animals

Our results demonstrated the protective properties of IL-37 on microglial activation, production of pro-inflammatory mediators, impairment of cognition, and disruption of long-term potentiation (LTP) after acute immunostimulatory challenge. To investigate the potential of IL-37 to attenuate chronic inflammation, we next analyzed transgenic APP/PS1 mice, which serve as a widely used animal model for Alzheimer's disease (*Jankowsky et al., 2004*), and crossed this mouse strain with hIL-37tg animals (*Figure 6A*; *Nold et al., 2010*).

First, we examined the levels of pro-inflammatory cytokines in the brains of 9–12 month-old control, APP/PS1 and APP/PS1-IL37tg transgenic mice (*Figure 6B–C*). Compared with control, APP/PS1 animals exhibited significantly higher IL-6 levels and a slight increase in IL-1β levels, although this was not statistically significant, whereas APP/PS1-IL37tg mice showed no increase in IL-6 and IL-1β levels. Moreover, the levels of these pro-inflammatory cytokines were significantly reduced in APP/PS1-IL37tg compared with APP/PS1 mice (IL-1β $F_{(2,19)}=5.224$ p=0.0156; p=0.0903; p=0.248; p=0.0046; IL-6 $F_{(2,21)}=3.6$ p=0.0452; p=0.022; p=0.5264; p=0.0466) (*Figure 6B–C*). We then examined microglial cell activation by FACS analysis. For this purpose, the percentage of microglial cells expressing the activation marker CD68 (*Jurga et al., 2020*; *Smith and Koch, 1987*; *Verbeek et al., 1995*) (identified as a CD11b+/CD45low cell population in the brain) was analyzed at different ages of mice (*Figure 6D–G*).

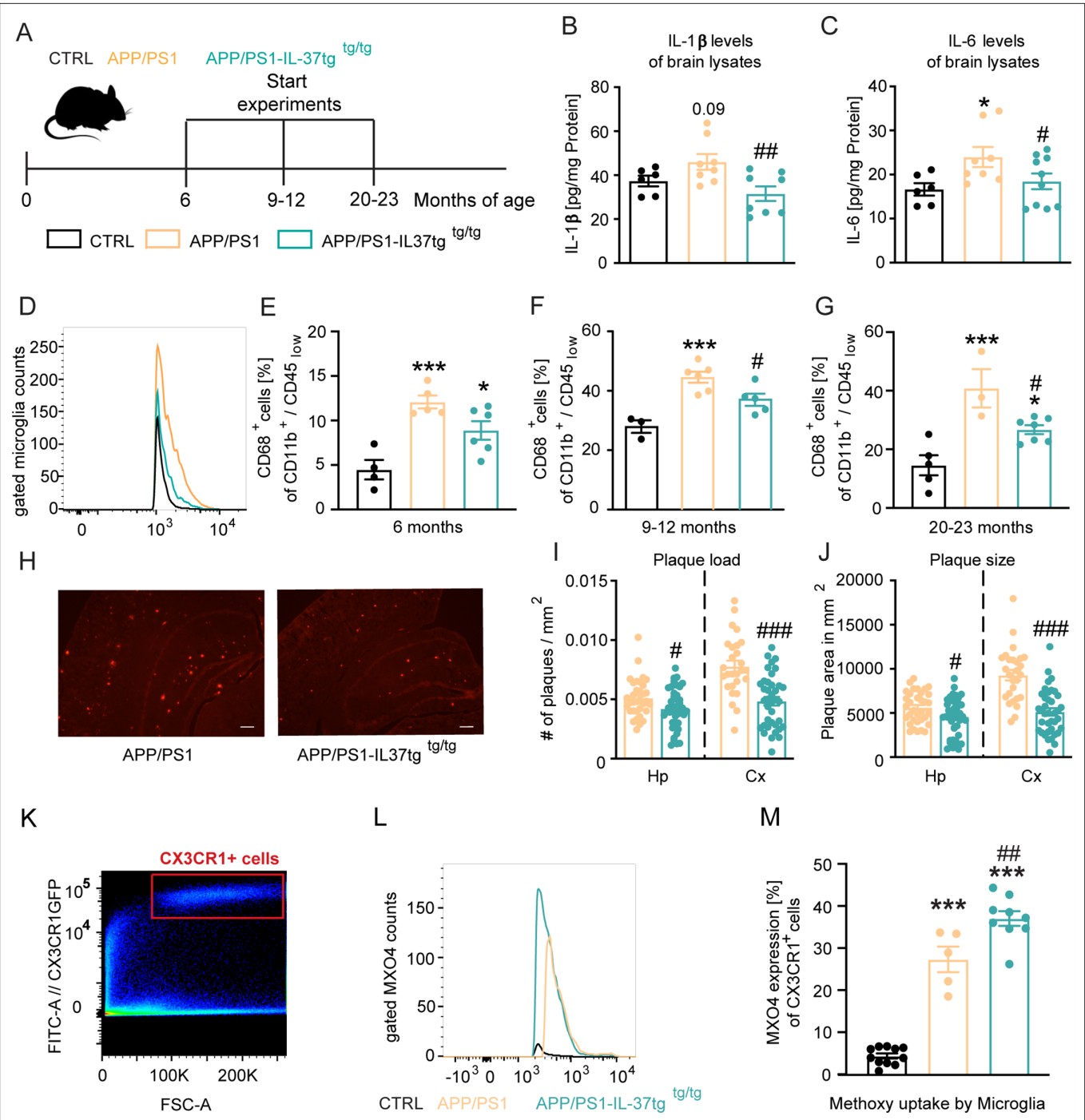

**Figure 6.** APP/PS1-IL37tg double transgenic mice showed lower pro-inflammatory cytokine expression and reduced activation of microglia, as well as lower numbers of amyloid plaques compared with APP/PS1 mice. (**A**) WT, APP/PS1, and APP /PS1-IL37tg mice were analyzed at 6, 9–12, and 20–23 months of age. (**B–C**) Pro-inflammatory cytokine levels of IL-6 and IL-1β (although for IL-1β was not statistically significant) were increased in 9–12 months old APP/PS1 mice compared with WT mice, whereas APP/PS1-IL37tg mice showed no increase in pro-inflammatory cytokines (n=6–10). (**D–G**) Microglial cells were identified as CD11b[+] and CD45[low] cells and analyzed for CD68 expression. Nine to 12-month-old APP/PS1 mice showed a significantly increased percentage of cells with CD68 expression compared with WT mice, whereas APP/PS1-IL37tg mice did not show a significant increase (**D and F**). Although 6- and 20–23 month-old APP/PS1-IL37tg mice had a significantly higher amount of CD68-expressing cells compared with WT mice, the percentage of CD68-expressing cells was reduced compared with APP/PS1 animals (6 months: n=4–6; 9–12 months: n=3–6; 20–23 months: n=3–7). (**H–J**) Plaque analysis showed significantly lower plaque burden (**I**) and reduced plaque size (**J**) in the hippocampal (Hp) and cortex (Cx) regions compared between APP/PS1-IL37tg mice and APP/PS1 (n=27–42). Representative image of Congo red staining in 30 μm sections; scale bar 500 μm (**H**). (**K–M**) Aβ uptake by microglial cells by measuring single cells in FACS system. Gating strategy for positive Cx3CR1-GFP cells and positive staining

*Figure 6 continued on next page*

*Figure 6 continued*

for MXO4 (**K–L**). Quantified analysis of Aβ uptake showing significantly higher uptake in APP/PS1-IL37tg cells compared with APP/PS1 cells (n=5–11) (**M**). Data are presented as mean ± SEM. * p<0.05, ** p<0.01, *** p<0.001 compared to WT, # p<0.05, ## p<0.01, ### p<0.001 compared to APP/PS1. (B-G and M: one-way ANOVA with multicolumn comparison; I-J: t-test).

The online version of this article includes the following source data and figure supplement(s) for figure 6:

**Source data 1.** Neuroinflammatory status in the brain of APP/PS1 mice.

**Figure supplement 1.** Congo red staining of brain sections from APP/PS1 and APP /PS1-IL37 mice.

**Figure supplement 2.** FACS analysis of microglial cells regarding their Methoxy-X04 uptake.

In the brains of 9–12 month-old animals, the frequency of CD68+ microglia isolated from APP/PS1 mice was significantly increased compared to age-matched WT, while the brains of APP/PS1-IL37tg mice did not show significantly increased numbers of CD68+ microglial cells (FACS 9–12 m $F_{(2,11)}$=14.39 p=0.0008; p=0.0008; p=0.0529; p=0.0494) (***Figure 6D and F***). The microglial cell population isolated from the brains of 6- and 20–23 month-old APP/PS1 and APP/PS1-IL37tg showed an increased proportion of CD68+ cells compared to those of WT controls (FACS 6 m $F_{(2,12)}$=13.45 p=0.0009; p=0.0007; p=0.0263; p=0.0985; FACS 20–23 m $F_{(2,12)}$=13.38 p=0.0009; p=0.0007; p=0.0359; p=0.0383) (***Figure 6E and G***). However, APP/PS1-IL37tg had a significantly lower percentage of cells expressing CD68 compared to APP/PS1 mice (***Figure 6F–G***). We then examined amyloid-β (Aβ) plaque load as a hallmark of Alzheimer's disease in the hippocampus and cortex (***Figure 6H–J***) of APP/PS1 (***Figure 6H*** left panel) and APP/PS1-IL37tg animals (***Figure 6H*** right panel). We observed that APP/PS1-IL37tg mice had significantly fewer plaques in both the hippocampus and cortex (***Figure 6I***) and that plaques were even smaller (***Figure 6J***) compared to APP/PS1 mice (plaque burden Hp t=2.403, df = 74 p=0.0187; Cx t=4.953, df = 63 p<0.0001; plaque size Hp t=2.296, df = 74 p=0.0245; Cx t=5.949, df = 63 p<0.0001) (***Figure 6H–J***; ***Figure 6—figure supplement 1A-D***).

To further analyze whether the observed lower Aβ plaques in APP/PS1-IL37tg were due to higher Aβ uptake, an Aβ uptake assay was performed to examine phagocytic activity in APP/PS1 and APP/PS1-IL37tg mice carrying the CX3CR1-GFP gene (heterozygous) (***Jung et al., 2000***). For this purpose, methoxy-XO4 staining was performed followed by FACS analysis. The CX3CR1-GFP transgenic mice express a microglia-specific green fluorescent protein in the CNS. To stain Aβ-plaques, animals were injected i.p. with 10 mg/kg methoxy-XO4 3 hr before the start of the experiment. After isolating the brain and performing a single-cell suspension, cells were measured in the FACS system and gated for Cx3CR1-positive cells (***Figure 6K***) and then further gated for the methoxy-XO4-positive population (***Figure 6—figure supplement 2A-I***). Data showed increased uptake of Aβ-particles by microglial cells from APP/PS1-IL37tg animals compared to those from APP/PS1 mice ($F_{(2,22)}$=136.2 p<0.0001; p<0.0001; p<0.0001; p=0.0026) (***Figure 6L–M***).

The APP/PS1 mouse model of Alzheimer's disease shows cognitive deficits as early as 8 months of age (***Jankowsky et al., 2004***; ***O'Leary and Brown, 2009***). To investigate whether IL-37 could positively affect learning and memory in this animal model, we performed the Morris water maze (MWM) behavioral test on 9–12 month-old control, APP/PS1, and APP/PS1-IL-37tg animals (***Figure 7A*** and ***Figure 7—figure supplement 1***). During the 8-day acquisition phase in the MWM, escape latency decreased progressively in all groups (***Figure 7B***). However, APP/PS1 animals showed increased escape latency on day 1 and day 3–6 of the training phase compared to control mice (escape latency $F_{(2,30)}$=10.5 p=0.0003; WT vs. APP/PS1 day 1 p=0.0463; day 3 p=0.007; day 4 p=0.0017; day5 p=0.0015; day 6 p=0.005) (***Figure 7B***). Subsequently, the reference memory test (probe trial) was performed on day 3 before the training and on day 9, 24 hr after the last training session. During the probe trials, the mice were tested without the presence of the escape platform. The percentage of time mice spent in each quadrant was measured, and preference for the target quadrant (TQ) was compared with the three non-target quadrants (NT). Control mice showed an explicit preference for the target quadrant at day 9 (t=11.45, df = 22 p<0.0001), whereas in the comparison APP/PS1 animals showed no preference for any of the quadrants (t=0.9874, df = 18 p=0.3365) (***Figure 7C***). Remarkably, APP/PS1-IL37tg mice showed significantly higher preference for the target quadrant, similar to what was observed in control animals (APP/PS1-IL37tg PT t=3.705, df = 20 p=0.0014) (PT $F_{(2,30)}$=8.415 p=0.0013; p=0.0009; p=0.0505) (***Figure 7C***). In addition, heat maps of the different groups (1 example per group) were shown to better represent the performance of the animals in the

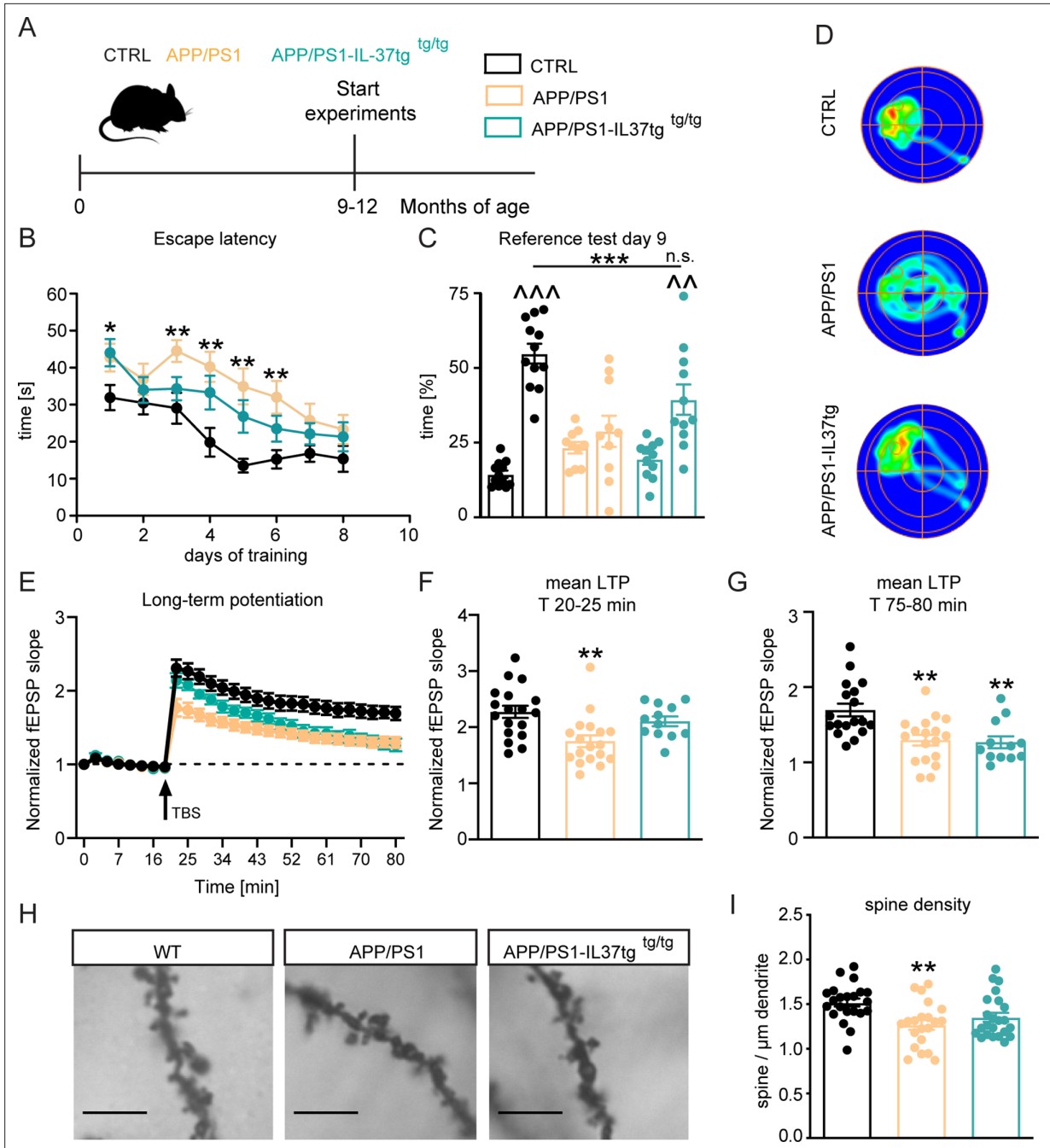

**Figure 7.** APP/PS1-IL37tg double transgenic mice showed improvements in behavioral tests and synaptic plasticity compared with APP/PS1 mice. (**A**) WT, APP/PS1 and APP/PS1-IL37tg mice were analyzed at 9–12 months of age. (**B–D**) The cognitive deficits of APP/PS1 mice in the spatial learning test of the Morris Water Maze could be restored in APP/PS1-IL37tg animals. WT APP/PS1 and APP/PS1-IL37tg mice showed learning behavior during the training phase of the spatial learning test. APP/PS1 animals showed higher escape latency during acquisition on day 3–6 compared to WT mice (**B**). WT Mice and APP/PS1-IL37tg mice show a significant preference for the target quadrant (TQ), whereas APP/PS1 mice showed no preference (**C**). Representative heat maps of mice from each group demonstrated the results of the reference test (**D**) (n=10–12). (**E–G**) The LTP deficits in APP/PS1 mice could be rescued in APP/PS1-IL37tg mice in the induction phase (20–25 min). However, LTP deficits were not restored in APP/PS1-IL37tg animals in the maintenance phase (75–80 min) (n of animals 3–4; n of slices 12–18). (**H–I**) Dendritic spine density was significantly reduced in APP/PS1 animals compared to WT, whereas there was no significant reduction in spine density in APP/PS1-IL37tg (n=21–23), scale bar 5 µm. Data are presented as mean ± SEM. * p<0.05, ** p<0.01, *** p<0.001 compared to WT, ^^ p<0.01, ^^^ p<0.001 compared to NT (non-target quadrants); (B+E: two-way ANOVA with multiple comparison; C: t-test; C-I: one-way ANOVA with multiple comparison).

The online version of this article includes the following source data and figure supplement(s) for figure 7:

*Figure 7 continued on next page*

*Figure 7 continued*

**Source data 1.** Assessment of spatial learning and synaptic plasticity in APP/PS1 mice.

**Figure supplement 1.** Age of the trained mice in the behavioral experiment.

**Figure supplement 1—source data 1.** Age range of mice trained in the Morris water maze.

reference memory test. These heat maps showed a prolonged time of control and APP/PS1-IL37tg mice in the target quadrant, whereas APP/PS1 mice did not show this preference (*Figure 7D*).

The observed improvement of learning and memory in APP/PS1-IL37 transgenic mice prompted us to analyze whether hippocampal network function could also be improved by this genotype. Therefore, we measured synaptic plasticity at the Schaffer collateral pathway as described above in these animals. Acute hippocampal slices from APP/PS1 mice showed significant deficits in LTP compared to corresponding slices from age-matched control mice (F(2,45)=9.286 p=0.0004) (*Figure 7E*). This was evident during both the induction (20–25 min of recording) and maintenance (75–80 min of recording) phases of LTP (first 5 min F(2,45)=7.09 p=0.0021; p=0.0016; last 5 min F(2,45)=9.354 p=0.0004; p=0.0014) (*Figure 7F–G*). Notably, the induction phase (but not the maintenance phase) of LTP was also indistinguishable in the slices from control and APP/PS1-IL37tg mice (first 5 min F(2,45)=7.09 p=0.0021; p=0.5444; last 5 min F(2,45)=9.354 p=0.0004; p=0.0023) (*Figure 7F–G*). Thus, it is clear that expression of IL-37 at 9 months of age in APP/PS1 mice can rescue the induction phase of LTP, which may be sufficient for the mice to perform spatial memory tasks.

To determine the rescued phenotypes observed in APP/PS1-IL37tg mice at the cellular level, neuronal morphology of the CA1 subregion of the hippocampus was analyzed in all experimental groups. A significant reduction in the density of dendritic spines was observed in 9–12 month-old APP/PS1 mice compared to age-matched control mice (dendritic spines F(2,63)=6.318 p=0.0032; p=0.0026) (*Figure 7H–I*). However, comparable dendritic spine density was detected between control and APP/PS1-IL37tg mice, indicating a rescue effect by IL-37 expression (dendritic spines F(2,63)=6.318 p=0.0032; p=0.0536) (*Figure 7H–I*). Taken together, these results suggest that expression of IL-37 in transgenic animals plays a protective role against chronic neuroinflammation by ameliorating the learning and memory deficits associated with the APP/PS1 mouse model and rescuing the underlying cellular correlates.

## RNA sequencing of microglia from IL-37tg mice reveals a slightly different gene expression profile after LPS challenge in vivo

Our results suggest that expression of IL-37 plays a protective role in both acute and chronic neuroinflammatory processes in the brain. Remarkably, in vitro experiments suggest that IL-37 exerts its anti-inflammatory effects most likely via microglial cells. To further investigate the function of microglia in this scenario, RNA sequencing was performed on microglial cells isolated from WT and IL-37tg mice treated with either saline or LPS. For this purpose, microglial cells were isolated using CD11b MicroBeads from WT and IL-37tg mice injected twice with saline or LPS 3 hr after the last injection, and then the mRNA expression profile of microglia was analyzed. The results showed that LPS challenge in vivo induced 11014 differentially expressed genes (DEGs) in microglial cells from WT and IL-37tg mice. The 500 most significantly expressed genes were shown in a heat map (*Figure 8A*). This highlights the fact that even acute systemic LPS exposure leads to tremendous changes in the gene expression profile specifically in microglia, suggesting a strong communication between the peripheral immune system and the brain. Gene set enrichment analysis (GSEA) showed that LPS stimulated genes of both genotypes associated with interleukin-1 signaling (*Figure 8B*). Although we did not detect significant changes in gene expression between WT and IL-37tg microglial cells after LPS stimulation (*Figure 8C*), some genes were identified that showed a slightly altered expression profile and have the potential to influence inflammatory outcome. Examples are: *Saa2*, which is highly expressed in response to inflammation and tissue injury (*Ye and Sun, 2015*; *Figure 8D*), *Vgf*, which is expressed after nerve injury and inflammation in neurons of both peripheral and central nervous systems and contributes significantly to the inflammatory processes, as blockade of VGF reduces the secretion of pro-inflammatory cytokines (*Busse et al., 2014*; *Figure 8E*), *Bcl6b*, which is involved in the regulation of inflammatory response and type 2 immune response (*Koyasu and Moro, 2011*; *Figure 8F*), *Il23a*, which is associated with autoimmune cholangitis and inflammatory bowel disease (*Ando et al.,*

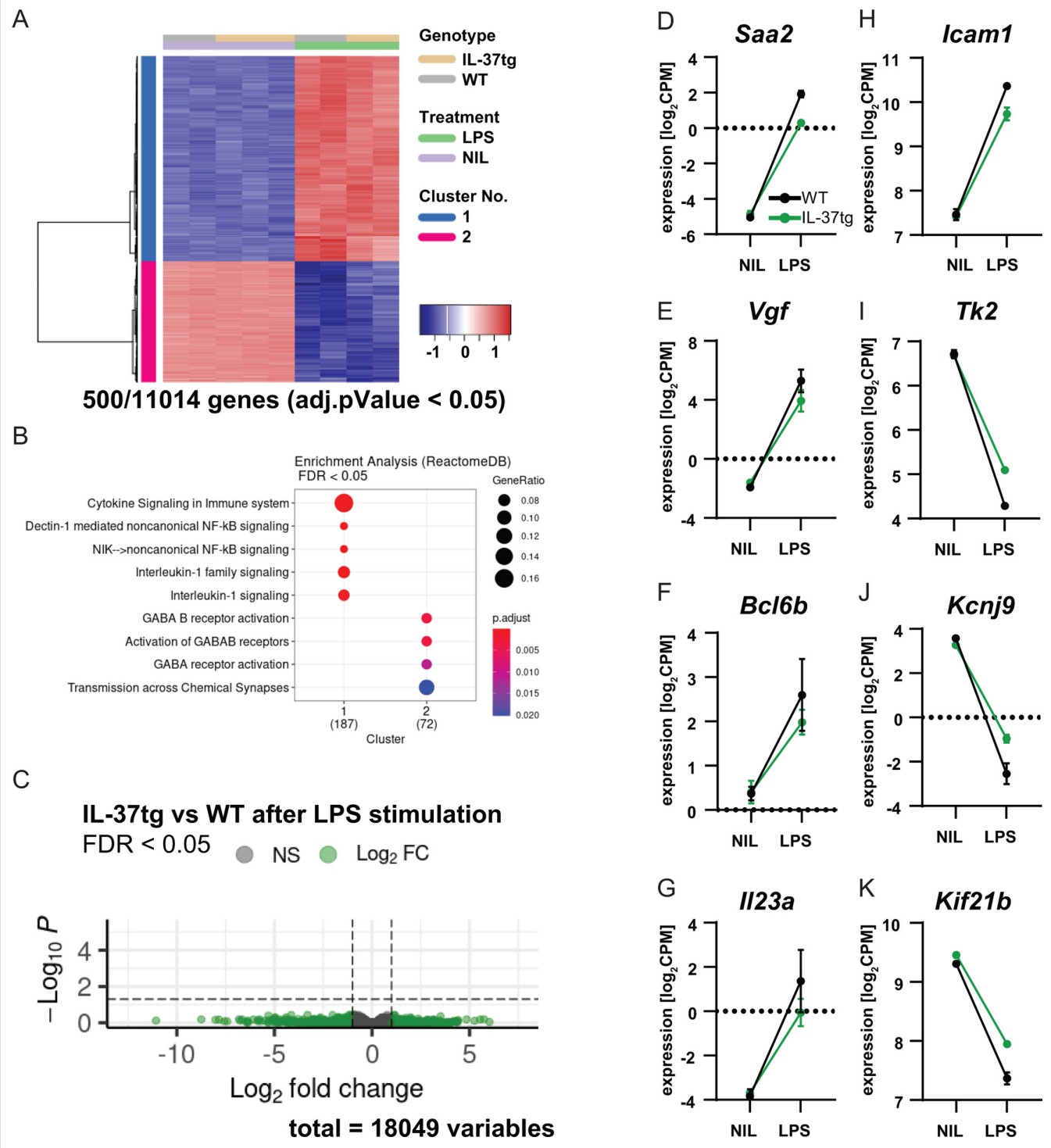

**Figure 8.** RNA sequencing of microglia isolated from WT and IL-37tg mice after LPS challenge in vivo. (**A**) Heatmap shows z-normalized gene expression profiles of the 500 most highly regulated genes (out of 11014 genes with adjusted pValue <0.05). Two clusters of co-regulated genes were identified, representing up- and down-regulated genes after LPS treatment. Each column shows expression data of individual mouse transcriptomes. (**B**) Gene Set Enrichment Analysis (GSEA) was performed for the 500 most up-regulated genes, which are also shown in the heat map. Each cluster (1 and 2) was tested for enriched gene sets defined by the Reactome Pathway Database (*https://reactome.org/*). The number of genes that could be linked to gene sets in the Reactome Pathway Database is indicated under each cluster name. The terms for the major gene sets are shown. The size of the circles corresponds to the ratio of genes found in each gene set. Significance of enrichment is expressed by a blue-red color code. (**C**) Differential expression comparing WT and IL-37tg after LPS stimulation is shown by the Volcano plot. Log₂FC and the corresponding adjusted pValue are shown for each gene

*Figure 8 continued on next page*

*Figure 8 continued*

analyzed (18,049 in total). The dashed lines show the limits for significant gene regulation: –1 1, FDR <0.05. (**D–K**) Changes in gene expression after LPS injection in WT and IL-37tg mice for candidate genes are shown (n=2-3).

The online version of this article includes the following source data and figure supplement(s) for figure 8:

**Source data 1.** RNA sequencing data of microglia isolated from mice after systemic LPS challenge.

**Figure supplement 1.** RNA sequencing of microglia isolated from WT and IL-37tg mice after LPS challenge in vivo.

**Figure supplement 1—source data 1.** Expression levels of candidate genes in microglia isolated from mice after systemic LPS challenge.

*2012*; *Figure 8G*), and *Icam1*, which plays a critical role in inflammatory processes and in the T-cell-mediated host defense system (*van de Stolpe and van der Saag, 1996*; *Figure 8H*). These genes were upregulated in WT mice after LPS administration, whereas they were differentially expressed to a lesser extent in IL-37tg mice treated with LPS. In addition, some genes were downregulated in the microglial cells of WT mice challenged with LPS, which was the case to a lesser extent in IL-37tg mice. For example, the *Tk2* gene, the deficiency of which is highlighted in mitochondrial depletion syndrome (*Zhou et al., 2013*; *Figure 8I*). The *Kcnj9* gene, which encodes the $Na^+/K^+$-ATPase pump and is important for brain function. Interestingly, this gene is significantly downregulated in schizophrenia (*Liu et al., 2019*; *Figure 8J*). The *Kif21b* gene, a kinesin protein that promotes intracellular transport and controls microtubule dynamics. Downregulation of this gene results in neurodevelopmental abnormalities due to imbalanced canonical motor activity (*Asselin et al., 2020*; *Figure 8K*). In addition to the above genes, some other important genes were equally regulated in both genotypes after LPS challenge, such as *Gpr84*, known to be regulated by pro-inflammatory cytokines such as TNF-α or IL-1 (*Bouchard et al., 2007*; *Figure 8—figure supplement 1A*), *Mmp3*, which is associated with brain inflammation and microglial activation (*Kim et al., 2005*; *Figure 8—figure supplement 1B*), *Acod1*, as a key regulator of immune metabolism in infection and inflammation (*Wu et al., 2020*; *Figure 8—figure supplement 1C*), and the IL-18 receptor (*Il18r*), indicating increased expression of the potential receptor for IL-37 in microglia after systemic LPS challenge (*Figure 8—figure supplement 1D*).

Although this mRNA expression assay for microglia showed that there were no significant differences between the overall gene profile of microglia from WT and IL-37 after LPS challenge, a specifically lower microglial response was nevertheless detectable in IL-37tg mice, which may shed light on the immunomodulatory role of IL-37 in acute and chronic neuroinflammation.

## Discussion

The regulation of inflammation is an extremely complex and tightly controlled signaling event in which cytokines crucially modulate the process (*Shaikh, 2011*). The interplay and balance between pro- and anti-inflammatory components are crucial for the delicate balance of inflammatory responses. Anti-inflammatory cytokines are known to limit persistent or excessive inflammation, but they may also be insufficient or overcompensated (*Shaikh, 2011*; *Kasai et al., 1997*; *Munoz et al., 1991*). IL-37 is expressed in many human tissues and cells, especially in monocytes, tissue macrophages or dendritic cells (*Cavalli and Dinarello, 2018*; *Su and Tao, 2021*; *Rudloff et al., 2017*), where it is produced in response to inflammatory stimuli and may act as a self-protective mechanism against escalating inflammatory processes (*Su and Tao, 2021*). In this study, we showed that IL-37, an anti-inflammatory cytokine, is able to reduce both acute and chronic neuroinflammation and that the IL-37tg mouse model is a valuable animal model to study the details of neurodegenerative diseases caused by chronic inflammation.

The acute inflammatory response was analyzed in a mouse model of septic shock in which increased levels of IL-6, TNF-α, and IL-1β produced by activated microglia were detected (*Lemstra et al., 2007*; *Widmann and Heneka, 2014*). The ability of IL-37 to suppress pro-inflammatory cytokines induced by Toll-like receptors (TLR) was first demonstrated in mouse macrophages transfected with human IL-37 (*Nold et al., 2010*) and recently described for TLR-induced activation of microglial cells (*Conti et al., 2020*). Because there is no homologous gene for IL-37 in mice, transgenic mice expressing the human IL-37 gene were generated (IL-37tg) (*Nold et al., 2010*; *Cavalli and Dinarello, 2018*). Here we showed the effect of IL-37 on CNS tissue resident macrophages (microglia) in terms of reducing inflammatory

markers similar to those in the periphery. In contrast, no reduction was detected in primary astrocytes from IL-37tg mice, suggesting a specific modulatory role of microglia in this scenario. The timing of LPS stimulation is important because prolonged treatment of 24 hr results in higher cytokine release compared to 6 hr (*Liu et al., 2018*; *Wu et al., 2009*).

Remarkably, the Warburg effect describes the change in metabolism toward the production of ATP from anaerobic glycolysis to oxidative phosphorylation observed in macrophages by increased mTOR levels and decreased AMPK activity after LPS stimulation (*Su and Tao, 2021*; *O'Neill and Hardie, 2013*). It was shown that IL-37 can reverse the Warburg effect (*Nold et al., 2010*; *Ballak et al., 2014*; *Nold-Petry et al., 2015*; *Su and Tao, 2021*) and, moreover, the metabolic cost of inflammation in plasma and muscle cells by reducing the concentration of succinate, a potent mediator in inflammatory states (*Tannahill et al., 2013*; *Cavalli et al., 2017*; *Fedotcheva et al., 2006*; *Mills and O'Neill, 2014*). In addition, macrophage activation has been found to increase the production of itaconate, which in turn inhibits succinate dehydrogenase (SDH), leading to an increase in succinate levels (*He et al., 2021*; *O'Neill and Artyomov, 2019*). Therefore, we focused here on two metabolites associated with inflammation in tissue macrophages. Our results showed that levels of itaconate and succinate decreased after LPS stimulation in primary microglia from IL-37tg animals. In conclusion, we demonstrated that the anti-inflammatory property of IL-37 is able to reduce inflammation in microglial cells in vitro. In addition, the restoration of metabolic cost plays a role in reversing cognitive decline in old age, as has been shown previously (*Minhas et al., 2021*).

To confirm the anti-inflammatory effect of IL-37 in acute inflammatory conditions in vivo, further studies were performed by systemic LPS injection in control and IL-37tg animals. Inflammation and activation of microglia are crucial hallmarks of various neurological diseases (*Qin et al., 2007*). On the one hand, activation of microglia is essential for homeostasis in the brain for an appropriate inflammatory response. On the other hand, overactivation of these processes can lead to neuronal damage (*Qin et al., 2007*; *McGeer et al., 2005*; *Polazzi and Contestabile, 2002*). In acute inflammation, little LPS is likely to enter the brain because the blood-brain barrier (BBB) is intact (*Nadeau and Rivest, 1999*). It is more likely that peripherally-induced neuroinflammation is an indirect effect (*Qin et al., 2007*), in which LPS activates the inflammatory cascade that signals to the CNS via TLR and/or TNF-α receptors on the BBB (*Qin et al., 2007*; *Block and Hong, 2005*; *Chakravarty and Herkenham, 2005*; *Ji et al., 2008*; *Kim et al., 2000*; *Laflamme et al., 2003*; *Wu et al., 2011*; *Yang et al., 2013*). Systemic inflammation induces increased density and reactivity of microglial cells characterized by increased secretion of pro-inflammatory cytokines such as IL-1 (*Widmann and Heneka, 2014*), changes in the morphological shape (retraction of the processes) (*Yang et al., 2013*; *Kettenmann et al., 2011*), greatly increased levels of surface proteins such as CD68 (*Jurga et al., 2020*; *Yang et al., 2013*; *Kettenmann et al., 2011*), and increased density of IBA-1-positive cells (*Yang et al., 2013*). In this study, we demonstrated that peripheral induced inflammation resulted in significantly increased numbers of IBA-1-positive cells and increased levels of CD68 and IL-1β in the CNS of control animals. In contrast, IL-37tg mice showed anti-inflammatory effects and did not exhibit an activated microglial phenotype in these features. However, the typical morphological changes described as retraction of processes (*Yang et al., 2013*) were not observed in either genotype. IL-37 was originally described as a basic inhibitor of innate immunity with reduced pro-inflammatory cytokine levels in plasma (*Nold et al., 2010*) and particularly in peripheral organs and in diseases such as endotoxemia, spinal cord injury, colitis, myocardial ischemia, obesity, and metabolic syndrome (*Nold et al., 2010*; *Cavalli and Dinarello, 2018*; *Ballak et al., 2014*; *Coll-Miró et al., 2016*; *McNamee et al., 2011*; *Yousif et al., 2011*; *Nold-Petry et al., 2015*). Our results clearly show that in vivo neuroinflammatory processes are reduced in acutely inflammatory challenged mice after transgenic IL-37 expression. IL-37 is a dual-function protein that has both intracellular and extracellular properties (*Cavalli and Dinarello, 2018*). Intracellularly, IL-37 interacts with SMAD3, translocates to the nucleus and induces anti-inflammatory effects (*Amo-Aparicio et al., 2021*; *Sharma et al., 2008*). Extracellularly, IL-37 binds to the IL-18Rα and the co-receptor IL-1R8 (SIGGIR), resulting in a signaling cascade with anti-inflammatory effects (*Nold-Petry et al., 2015*; *Amo-Aparicio et al., 2021*). The differences between these two signaling pathways were demonstrated by *Amo-Aparicio et al., 2021* in a spinal cord injury model (*Amo-Aparicio et al., 2021*).

The resulting neuroinflammation and activation of microglial cells after peripheral LPS challenge also lead to functional and structural changes in the neuronal network, particularly affecting long-term

potentiation (LTP) (*Beyer et al., 2020*; *Commins et al., 2001*; *Di Filippo et al., 2013*; *Hennigan et al., 2007*; *Strehl et al., 2014*) and spine morphology with implications for dendritic spine density (*Beattie et al., 2002*; *Boulanger, 2009*; *Ikegaya et al., 2003*; *Kondo et al., 2011*; *Lucin and Wyss-Coray, 2009*). Here, we showed that LTP was impaired in control animals after immunostimulation with LPS, whereas IL-37tg animals showed no defects in LTP. Loss of dendritic spines after LPS challenge was also restored in IL-37tg stimulated with LPS. These data indicate that deleterious neuronal changes known to be associated with acute inflammatory processes in the CNS are reduced in IL-37tg mice.

Our results suggest decreased neuroinflammation induced by systemic immune stimulation in the presence of IL-37. However, the anti-inflammatory effect of IL-37 in the CNS may be a secondary effect due to the reduction of the inflammatory response in the periphery. To identify the direct possibility of IL-37 expression in the CNS without peripheral influence, further electrophysiological experiments were performed with acute hippocampal slices from control and IL-37tg mice. Administration of LPS to the acute slices of IL-37tg mice suggests that IL-37 expression directly in the CNS may also be responsible for the reversal of neuronal impairments after peripheral LPS challenge. In addition, we confirmed the mechanisms of anti-inflammatory action in IL-37tg mice by using the recombinant IL-37 in WT mice. Recombinant human IL-37 has protective effects against endotoxemia, acute lung and spinal cord injury, asthma, and myocardial infarction (*Cavalli et al., 2016*; *Cavalli and Dinarello, 2018*; *Coll-Miró et al., 2016*; *Moretti et al., 2014*; *Wu et al., 2014*; *Li et al., 2015*; *Lunding et al., 2015*; *Ye et al., 2014*). Next, we investigated whether recombinant IL-37 has a similar effect on inflammatory processes in the CNS. Our results showed a reduction in pro-inflammatory cytokine levels in the brain. The consequences of acute peripheral immune stimulation by either LPS or IL-1β as a key cytokine triggering innate immune responses are the induction of pro-inflammatory cytokines leading to impaired learning and memory (*Arai et al., 2001*; *Gonzalez et al., 2013*; *Rachal Pugh et al., 2001*; *Shaw et al., 2001*; *Sparkman et al., 2005*; *Yirmiya and Goshen, 2011*). The effect of reduced neuroinflammation was analyzed on cognition and demonstrated by behavioral improvements in the Y-Maze test in animals pretreated with rIL-37. Spontaneous alternation behavior (SAB) in a symmetrical Y-shaped maze was used previously for pharmacological studies of short-term memory performance (*Drew et al., 1973*; *Hughes, 2004*; *Kokkinidis and Anisman, 1976*; *Swonger and Rech, 1972*). Systemic inflammation primarily affects attention and cognitive flexibility, including working memory, rather than associative learning. Low-dose injection of LPS, which elicits a very mild immune response and small changes (<1 °C) in core body temperature, has been shown to severely impair working memory (*Murray et al., 2012*). We therefore show that systemic injection of rIL-1β completely abolishes working memory. Consequently, rIL-37 was used as a preventive strategy by generating a certain level of IL-37 in the WT mice. Other studies using rIL-37 as a therapeutic intervention were performed, for example, after spinal cord injury by local administration of rIL-37 (*Coll-Miró et al., 2016*; *Amo-Aparicio et al., 2021*).

In this study, we specifically analyzed the activation status of microglial cells as a neuroinflammatory feature leading to structural changes in the neuronal network. Although our data suggest that IL-37 most likely transmits its signals via microglia, astrocytes also play an important role in maintaining neuronal circuits (*Nizami et al., 2019*; *Schafer et al., 2012*). Activated astrocytes express many complement factors that lead to synapse formation and elimination (*Nizami et al., 2019*; *Stephan et al., 2012*; *Stevens et al., 2007*). Therefore, it would be of interest in future experiments to dissect the underlying mechanism of synapse elimination with respect to IL-37 expression including microglia and astrocytes.

To better predict the role of microglia in IL-37 signal transduction after immunostimulation, single-cell type RNA sequencing was performed. Analysis of mRNA gene expression profiles revealed that approximately 11014 genes were differentially expressed in microglia after systemic LPS administration, clearly confirming the close communication between the peripheral immune system and the CNS. Although no significant differences were found between the microglial gene profile of WT and IL-37tg mice, however, these data indicated the specific gene profile in the both genotypes after the LPS challenge. For example, some genes were up- or down-regulated in the WT mice, which was the case to a lesser extent in the IL-37tg mice. Among them were some important genes such as Saa2 (*Ye and Sun, 2015*), Vgf (*Busse et al., 2014*), Bcl6b (*Koyasu and Moro, 2011*), Il23a (*Ando et al., 2012*), Icam1 (*van de Stolpe and van der Saag, 1996*), and Tk2 (*Zhou et al., 2013*), which contribute to inflammatory responses and are involved in many inflammation-related diseases. Interestingly,

peripheral LPS exposure led to downregulation of some genes such as Kcnj9 (*Liu et al., 2019*) and Kif21b (*Asselin et al., 2020*), the deficiency of which has been associated with the occurrence of neurological and neurodevelopmental disorders, again highlighting the importance of the immune system-brain axis. Moreover, peripheral LPS challenge induced the upregulation of some genes equally in both genotypes, such as *Gpr84* (*Bouchard et al., 2007*), *Mmp3* (*Kim et al., 2005*) which are associated with neuroinflammation and neurodegeneration. Or *Acod1*, which produces the immune metabolite itaconate that has been shown to be upregulated after LPS administration and to inhibit inflammatory signaling in monocytes (*Wu et al., 2020*). Our mRNA gene expression analysis showed upregulation of *Acod1*, suggesting a link between the in vitro model (primary microglia) and the in vivo model here, because primary microglia showed upregulation of itaconate upon LPS treatment. Finally, RNA sequencing data showed upregulated gene expression of the IL-18 receptor, suggesting that expression of the IL-18 receptor on microglial cells may be responsible for IL-37 signal transduction by microglia in IL-37tg mice. It is worth noting that the lack of large differences in microglial gene expression profiles between WT and IL-37tg mice may be due to the timing of microglial isolation, because we isolated microglia 3 hr after the last LPS injection, which was also the case for the other experiments. However, the timing of gene regulation is different from that of protein synthesis. Therefore, it can be assumed that 3 hr after the last LPS injection, when we detected the deleterious effects, the transcripts of the responsible genes have already been degraded.

In the first part of this study, we showed that acute neuroinflammation, microglial activation, deficits in functional and structural plasticity, and impaired cognition induced by peripheral immune stimulation were attenuated or rescued in the presence of IL-37 (either transgene or recombinant). Moreover, these results suggest that rIL-37 may play an important future role as a therapeutic intervention in many neurological diseases associated with inflammatory responses. However, further experiments with recombinant IL-37 need to be performed to obtain a detailed picture of the mechanism underlying rIL-37 signaling. A recent study has shown that the efficacy of IL-37 is limited when administered systemically, suggesting a beneficial effect on local CNS cells rather than cells in the periphery (*Amo-Aparicio et al., 2021*). Further experiments need to analyze therapeutic approaches for acute neuroinflammation, as it is not clear whether treating the peripheral immune response is sufficient to produce beneficial effects in brain tissue. Here, we show that local expression of IL-37 in the brain is sufficient to rescue LPS-induced LTP impairments. However, translation to brain impairments in acute systemic inflammation remains to be investigated with respect to systemic rIL-37 administration.

To analyze chronic neuroinflammation, we used APP/PS1 mice as a model for Alzheimer's disease (AD). AD is the most common form of dementia, and in 2015, an estimated 46.8 million people worldwide were living with the dementia (*Prince, 2015*). Research on AD has mainly focused on the long-lasting known pathological hallmarks namely beta-amyloid plaques and neurofibrillary tangles (*Ardura-Fabregat et al., 2017*; *Glabe, 2005*; *Hardy and Selkoe, 2002*; *Wang et al., 2015*). Recently, reports linking neuroinflammation to the pathogenic process of AD have been accumulating, showing that the brain can no longer be considered as an absolutely immune-privileged organ in disease progression (*Ardura-Fabregat et al., 2017*; *Wang et al., 2015*; *Heneka et al., 2015*; *Heneka et al., 2014*). In addition to the accumulation of amyloid-β (Aβ) during the progression of neuroinflammatory processes, pro-inflammatory cytokines such as TNF-α, IL-1β, or IL-6 have been shown to promote phosphorylation of tau (*Bhaskar et al., 2010*; *Kitazawa et al., 2011*; *Quintanilla et al., 2004*; *Xu et al., 2021*). In the present study, we focused on reducing the chronic neuroinflammatory response by decreasing microglial cell activation and proinflammatory mediators release and the subsequent effect on Aβ accumulation. For this purpose, we used the IL-37tg mouse line described above. To investigate the role of the anti-inflammatory cytokine IL-37 in chronic inflammation, we crossed the IL-37tg mouse with the APP/PS1 mouse. The APP/PS1 mouse line represents a reliable model that exhibits Aβ deposition as early as 6 months of age and has been reported to exhibit cognitive deficits starting at 8 months of age (*Jankowsky et al., 2004*; *O'Leary and Brown, 2009*; *Garcia-Alloza et al., 2006*). Microglial cells, the resident immune cells of the brain, are described (along with astrocytes) as a major source of cytokines that have a significant and distinct impact on the neuroinflammation aspects in AD (*Heneka et al., 2015*; *Prinz et al., 2011*). It has been previously described that microglia respond to Aβ peptides, which might be able to trigger the inflammatory process in AD that contributes to microglial activation, release of pro-inflammatory cytokines and memory deficits (*Wang et al., 2015*; *Heneka et al., 2015*; *Zilka et al., 2012*). Here, we show reduced pro-inflammatory cytokine

levels for IL-1 and IL-6 in brain lysates from 9- to 12-month-old APP/PS1-IL37tg mice compared to APP/PS1 animals. In addition, microglial activation was significantly higher in APP/PS1 mice compared to control mice, whereas APP/PS1-IL37tg mice did not show a significant increase in microglial activation due to CD68 expression. Furthermore, we analyzed Aβ deposition in the brain tissue of APP/PS1 animals compared with APP/PS1-IL37tg mice. Overexpression of the anti-inflammatory cytokine IL-37 in APP/PS1 animals used in this study resulted in significantly higher plaque phagocytosis capacity, lower plaque burden, and smaller plaque size compared to APP/PS1 control mice. In conclusion, reduction of pro-inflammatory response or inhibition of excessive inflammation by IL-37 expression may lead to better Aβ uptake and may be involved in higher clearance of Aβ plaques.

It is noteworthy that previous studies have shown that IL-1β and IL-6 play an important role in the progression of AD, as both were released from microglial cells surrounding Aβ plaques in AD patients and animal models (*Wang et al., 2015*; *Boutajangout and Wisniewski, 2013*; *Hunter et al., 2012*; *Vukic et al., 2009*). Furthermore, IL-1β and IL-6 are described to play complex roles in regulating cognitive function in AD (*Yirmiya and Goshen, 2011*; *Wang et al., 2015*; *Kitazawa et al., 2011*; *Dugan et al., 2009*; *Weaver et al., 2002*). The present results showed an impairment of spatial memory in 9- to 12-month-old APP/PS1 mice compared with control animals. However, APP/PS1-IL37tg mice showed partially rescued spatial memory performance compared with APP/PS1 mice. In addition, APP/PS1-IL37tg mice did not exhibit any impairment during the reference memory test. Given the observed impairment in spatial learning in APP/PS1 animals, we further analyzed synaptic plasticity by measuring LTP as a cellular correlate of learning and memory in all groups. The findings demonstrated that the maintenance and induction phases of LTP were impaired in APP/PS1 mice compared to control animals. In contrast, the induction phase of LTP in APP/PS1-IL37tg mice was similar to that in control animals. However, the maintenance phase of LTP was also impaired in APP/PS1-IL37tg animals. These observations suggest that overexpression of IL-37 is sufficient to rescue the early phase of LTP (E-LTP; the first 20 min after stimulation), which is independent of protein synthesis. However, impaired late LTP in APP/PS1 mice, which requires regulation of gene expression at the transcriptional and translational levels (*Auffret et al., 2010*), was not rescued by IL-37 overexpression. Interestingly, there is evidence that the water maze task can be efficiently solved despite late-LTP impairments in the CA1 or the CA3 hippocampal subregion (*Bannerman et al., 2012*; *Nakazawa et al., 2002*). In addition, neuronal morphology was analyzed to investigate whether structural cellular changes were the reason for the decline in LTP and cognitive function. When APP/PS1 mice were compared with control mice, it was found that APP/PS1 mice exhibited decreased spine density in the apical dendrites of CA1 pyramidal neurons. In contrast, APP/PS1-IL37tg did not show decreased numbers of dendritic spines.

In general, the putative mechanism of IL-37 signaling in neurodegenerative diseases is not clear. Translation to humans might suggest IL-37 as a biomarker for inflammatory diseases with changes in IL-37 levels in patients associated with these diseases (*Su and Tao, 2021*). IL-37 as a dual-acting cytokine during AD progression, we suggest that both intracellular and extracellular IL-37 signaling can reduce neuroinflammation and microglial activation. Because IL-37 is also produced by peripheral cells, this could lead to a reduced inflammatory response and a reduction in activated circulating immune cells, whether or not these peripheral immune cells enter the brain. In addition, it is not clear whether infiltrating immune cells have positive or negative effects on AD progression. There are some studies showing that phagocytic peripheral immune cells can infiltrate the brain and serve as a surrogate for microglial populations (*Dionisio-Santos et al., 2019*; *Fiala et al., 2002*; *Merlini et al., 2018*; *Thériault et al., 2015*; *Town et al., 2005*).

In summary, during chronic neuroinflammation associated with Alzheimer's disease, a significant deficit in synaptic plasticity (functional LTP and structural spine density) was documented, whereas rescue was observed in transgenic AD animals overexpressing the anti-inflammatory cytokine IL-37. In addition, a significant effect on the formation of amyloid-β plaques, a pathological feature of AD, was documented, suggesting that attenuation of neuroinflammation combined with increased clearance is sufficient to produce beneficial effects on learning and memory. Indeed, the results reported here demonstrate that expression of an anti-inflammatory IL-37 cytokine is able to reduce neuroinflammation and cognitive decline in a mouse model of Alzheimer's disease. Furthermore, recent studies in spinal cord injury (*Coll-Miró et al., 2016*; *Amo-Aparicio et al., 2021*) and in a mouse model of multiple sclerosis (MS) *Cavalli et al., 2019*; *Sánchez-Fernández et al., 2021* have shown that IL-37

signaling has protective properties in two other neuroinflammatory models by both transgenic expression and recombinant delivery (*Su and Tao, 2021*). Overall, the results of this study highlight the role of IL-37 in alleviating acute and chronic neuroinflammatory conditions and provide the basis for recombinant IL-37 as a potential future therapeutic approach. Furthermore, the IL-37tg mouse is a novel and important model system to explore therapeutic options while gaining mechanistic insights into human neurodegenerative diseases.

# Materials and methods

## Key resources table

| Reagent type (species) or resource | Designation | Source or reference | Identifiers | Additional information |
|---|---|---|---|---|
| antibody | Anti-IBA1 (Rabbit polyclonal) | Synaptic Systems | Cat#234003, RRID:AB_10641962 | 1:1000 |
| antibody | Cy2 AffiniPure Goat Anti-Rabbit IgG (H+L) (Rabbit polyclonal) | Jackson ImmunoResearch Laboratories | Cat# 111-225-144, RRID:AB_2338021 | 1:500 |
| antibody | Cy3 AffiniPure Goat Anti-Mouse IgG +IgM (H+L) (Mouse polyclonal) | Jackson ImmunoResearch Laboratories | Cat#115-165-068, RRID:AB_2338686 | 1:500 |
| antibody | Cy3 AffiniPure Goat Anti-Rabbit IgG (H+L) (Rabbit polyclonal) | Jackson ImmunoResearch Laboratories | Cat#111-165-144, RRID:AB_2338006 | 1:500 |
| antibody | mouse CD68-PE Clone REA835 (mouse monoclonal) | Miltenyi Biotec | Cat# 130-112-856 | 1:50 |
| antibody | mouse CD11b-PerCP-Vio700 Clone REA592 (mouse monoclonal) | Miltenyi Biotec | Cat# 130-113-809 | 1:50 |
| antibody | mouse CD45-APC (mouse monoclonal) | Miltenyi Biotec | Cat# 130-110-798 | 1:50 |
| chemical compound, drug | Bovine Serum Albumin | Sigma-Aldrich | Cat# A7906 | |
| chemical compound, drug | $CaCl_2$ | Applichem | Lot: 4U010421 | |
| chemical compound, drug | cOmplete Protease Inhibitor Cocktail | Sigma-Aldrich | Cat# 04693116001 | |
| chemical compound, drug | DAPI | Sigma-Aldrich | Cat# D9542 | |
| chemical compound, drug | D-glucose | Roth | Art.-Nr. HN06.3 | |
| chemical compound, drug | Evans Blue tetrasodium salt | Tocris | Cat# 0845 | |
| chemical compound, drug | Fluoro-Gel mounting medium | Electron Microscopy Sciences | Cat# 17985–10 | |
| chemical compound, drug | GBSS | Sigma-Aldrich | G9779-500ML | |
| chemical compound, drug | Gibco DMEM | Fisher Scientific | Cat# 31885023 | |
| chemical compound, drug | Gibco Fetal Bovine Serum | Fisher Scientific | Cat# 11573397 | |
| chemical compound, drug | Gibco HBSS 10 X | Fisher Scientific | Cat# 14065049 | |
| chemical compound, drug | Gibco L-Glutamine | Fisher Scientific | Cat# 15410314 | |
| chemical compound, drug | Glycine | Applichem | Cat# A1067 | |
| chemical compound, drug | KCl | Applichem | Lot: 0000574737 | |

*Continued on next page*

*Continued*

| Reagent type (species) or resource | Designation | Source or reference | Identifiers | Additional information |
|---|---|---|---|---|
| chemical compound, drug | KH$_2$PO$_4$ | Applichem | Lot: 4Q016683 | |
| chemical compound, drug | Methoxy-XO4 | Abcam | ab142818 | |
| chemical compound, drug | MgSO$_4$ | Applichem | Lot: 3E000057 | |
| chemical compound, drug | NaCl | Applichem | Lot: 8Q012497 | |
| chemical compound, drug | NaHCO$_3$ | Roth | Art.-Nr. HN01.1 | |
| chemical compound, drug | Permount Mounting Medium | Fisher Scientific | Cat# SP15-100 | |
| chemical compound, drug | Poly-L-lysine solution | Sigma-Aldrich | CAS# 25988-63-0 | |
| peptide, recombinant protein | Recombinant IL-37 | *Moretti et al., 2014* | N/A | |
| peptide, recombinant protein | Recombinant IL-1β | *Kim et al., 2013* | N/A | |
| chemical compound, drug | Triton X-100 Molecular Biology grade BC | Applichem | Cat# A4975 | |
| chemical compound, drug | Trypsin-EDTA Solution 10 X | Sigma-Aldrich | CAS Nr. 9002-07-7 | |
| chemical compound, drug | TWEEN 20 | Sigma-Aldrich | Cat# P9416 | |
| commercial assay or kit | Biozym Blue Probe qPCR Kit Separate ROX | Biozym | Cat# 331456 S | |
| commercial assay or kit | FD Congo-Red Solution Kit | FD NeuroTechnologies, Inc. | Cat# PS108 | |
| commercial assay or kit | FD Rapid GolgiStain Kit | FD NeuroTechnologies, Inc. | Cat# PK401 | |
| commercial assay or kit | High-Capacity cDNA Reverse Transcription Kit | Thermo Fisher | Cat# 4368814 | |
| commercial assay or kit | Macherey-Nagel NucleoSpin RNA | Thermo Fisher | Product Code 15373604 | |
| commercial assay or kit | pegGOLD TriFast | Avantor | N/A | |
| commercial assay or kit | ProtoScript II First Strand cDNA Synthesis Kit | New England Biolabs Inc. | Cat# E6560 | |
| strain, strain background (*Mus musculus*) | B6;C3-Tg(APPswe,PSEN1dE9)85Dbo/Mmjax mice | The Jackson Laboratory | Cat# 005864 | |
| strain, strain background (*Mus musculus*) | C57BL/6 J OlaHsd mice | Harlan-Winkelmann or Janvier | Cat# 057 (H-W) | |
| strain, strain background (*Mus musculus*) | Human Interleukin-37 transgenic mice | *Nold et al., 2010* | N/A | |

*Continued on next page*

*Continued*

| Reagent type (species) or resource | Designation | Source or reference | Identifiers | Additional information |
|---|---|---|---|---|
| software, algorithm | ANY-maze | Stoelting | RRID:SCR_014289 https://www.stoeltingco.com/ | |
| software, algorithm | FlowJo | FlowJo | https://www.flowjo.com/solutions/flowjo | |
| software, algorithm | ImageJ | Wane Rasband NIH, USA | N/A | |
| software, algorithm | IntraCell Version 1.5 | (C)2000 Institute for Neurobiology Magdeburg | N/A | |
| software, algorithm | Prism 8 | GraphPad | https://www.graphpad.com/scientific-software/prism/ | |
| software, algorithm | Video Mot 2 | TSE Systems | https://www.tse-systems.com | |
| software, algorithm | G*Power Version 3.1.9.4 | Heinrich Heine University Düsseldorf, Germany | http://www.psychologie.hhu.de/arbeitsgruppen/allgemeine-psychologie-und-arbeitspsychologie/gpower.html | |

All experiments were performed and analyzed blinded to the experimenter.

## Animals

All animals used in this study were of either sex, equal distributed to the experiments with exception in the behavioral test were only males were used (*Figure 7—figure supplement 1*). Mice were bred and kept at the animal facility of the TU Braunschweig under standard housing conditions in a 12:12 light:dark cycle at 22 °C with food and water available ad libitum. Transgenic mouse expressing human IL-37 (hIL-37tg mice, also referred as IL-37tg) were provided by Prof. Dr. Philip Bufler, Medical University of Munich. IL-37tg animals were originally generated by injecting fertilized eggs from C57BL/6 females with the pIRES IL-37 expression plasmid (*Nold et al., 2010*). As a starting pair for these experiments here, a heterozygous female was mated with a heterozygous male. Resulting negative animals, heterozygous and homozygous IL-37tg animals were identified by PCR. The distinction between heterozygous and homozygous animals could be clearly analyzed based on the positive mRNA quantity. Further C57BL/6 J wild-type (WT) mice and APP/PS1ΔE9 (heterozygous breeding) mice were used. The latter mouse line was crossed with the hIL-37 line to create a double-transgenic mouse line (APP/PS1-IL37). In all experiments negative littermates and/or C57BL/6 wild-type mice were used as controls. All experimental procedures were authorized by the animal welfare representative of the TU Braunschweig and the LAVES (Oldenburg, Germany) (33.19-42502-04-16/2170).

## Lipopolysaccharide (LPS) administration

In all performed experiments related to any lipopolysaccharide (LPS) stimulus, the same LPS from *Escherichia coli* (*E. coli* O127:B8, Sigma Aldrich L 3129) was used. The systemic immune stimulation with LPS was performed by intraperitoneal injection. The body weight of the stimulated animals was monitored to determine the appropriate volume of LPS. Animals were either injected with 2x0.5 mg/kg LPS or 0.9% sodium chloride (NaCl) as a control.

## LPS administration during electrophysiological recordings

To investigate the acute effect of LPS-induced direct immunostimulation and subsequent local IL-37 expression in the CNS, LPS administration was performed during electrophysiological recordings. For this purpose, acute hippocampal slices from adult control and IL-37tg mice were pretreated with LPS (10 µg/ml) 2 hr before and throughout the recording period.

## Administration of recombinant IL-37 together with either IL-1β or LPS

Wild-type mice were injected intraperitoneally with 300 ng of recombinant IL-37 per mouse for 3 consecutive days, followed by an injection of 60 ng of recombinant IL-1β per mouse.

To test the preventive effect of rIL37 with LPS, WT mice received an intraperitoneal injection of 100 ng of rIL-37 or vehicle for three consecutive days. Two hours after the third injection, mice received an injection of either 0.5 mg/kg LPS or 0.9% sodium chloride (NaCl) as a control, followed by a second injection 24 hr later.

## Cell culture and LPS administration

Neonatal mouse brains (P3 – P5) were used for culture preparation as shown previously (*Lonnemann et al., 2020*). Briefly, the meninges were removed and the brain transferred into HBSS 1 X on ice. Using a 10 ml pipette, the tissue was transferred into a sterile 50 ml conical tube and centrifuged at 2000 rcf for 5 min at 4 °C. The re-suspended pellet (in 5 ml fresh HBSS 1 X) was applied on a cell strainer (100 µm pores). Again after spinning as before the pellet was re-suspended in 10 ml culture media (DMEM +10% FCS+1% Penicillin/Streptomycin) and transferred into a T-75 flask. The mixed culture was incubated in the flask in an incubator at 10% $CO_2$ at 37 °C for 2–3 weeks. After 3 days incubation, the media was replaced 50% with fresh media. In the following every 7 days the media was replaced completely. After 2–3 weeks, the culture has reached confluence and the flasks were shaken at 180 rpm for 3 hr at 37 °C. The media including the microglia cells was collected without disrupting the astrocyte layer on the bottom of the flask and was centrifuged at 3000 rpm for 10 min at room temperature (RT). Microglia cells were plated in 96-well plate with a density of $6x10^4$ cells/ well and were treated with different concentrations of LPS for 24 hr. In the last hour of the treatment, ATP (5 mM) was added to the cells.

To prepare a primary astrocyte culture, neonatal mouse brains (P3-4) were used. After removal of the brain, the hippocampus and meninges were carefully removed. The cortices were transferred on ice to fresh HBSS 1 x, and the tissue was homogenized using a 10 ml pipette. The HBSS was then washed off, and the brains were placed in a dissociation solution containing DNAse for 30 min at 37 °C, and the remaining tissue pieces were further dissociated by pipetting. After centrifugation at 800 rpm for 7 min, the supernatant was removed and the cells were resuspended in culture medium (DMEM supplemented with 10% FCS, 1% penicillin/streptomycin) and placed on a 40 µm cell strainer and finally transferred to a T-75 flask coated with poly-D-lysine. When cells reached confluence, they were shaken overnight at 220 rpm to remove other glial cells, and then astrocytes were passaged. The cells were passaged three times until the experiments were performed. Experiments were performed on day in vitro (DIV) 15–19. Astrocytes were plated in a 12-well plate ($1×10^5$ cells/well) and maintained for 24 hr followed by a 100 ng/ml LPS stimulus.

## Cytokine measurement of pro-inflammatory IL-6, IL-1β, and TNF-α

Enzyme-linked immunosorbent assay (ELISA) was used to quantify cytokines in either brain homogenates or supernatants of treated primary microglia cells. Mice were deeply anesthetized with $CO_2$ and killed via decapitation. Brains were isolated and homogenized in STKM buffer (250 mM Sucrose, 50 mM Tris-HCl, 25 mM KCl, 5 mM $MgCl_2$) using the GentleMACS (Miltenyi Biotec) program Protein_01. After centrifugation at 4000 g for 5 min at 4 °C the supernatant was again centrifuged for 10 min at 13,000 g at 4 °C. Brain homogenates (1:2) (diluted in 1% BSA solution) were analyzed using R&D systems ELISA Kits.

## IL-37 mRNA measurement

To quantify the expression of IL-37 in primary microglia, cells were treated with 100 ng/mL LPS for 24 hr for seven different time periods (5 Min, 10 Min, 20 Min, 40 Min, 1 hr, 4 hr and 24 hr). Total RNA was isolated using peqGOLD TriFast. cDNA was prepared using BioLabs ProtoScript II First Strand cDNA Synthesis Kit.

For tissue extraction the RNA purification Kit (Macherey-Nagel) was used. cDNA was prepared by using the High Capacity cDNA Reverse Transcription Kit (Thermofisher).

Then, real-time quantitative PCR (qPCR) was performed using BlueProbe qPCR Mix (Biozym) and the following primer pairs: *IL-37*: 5′-GGG AGT TTT GTC TCT ACT GTG AC-3′(forward) and 5′-CCC ACC TGA GCC CTA TAA AAG-3′(reverse); *GAPDH* 5′-GCC TTC CGT GTT CCT ACC-3′(forward) and 5′-CCT CAG TGT AGC CCA AGA TG-3′(reverse). Expression levels of target mRNA was analyzed using the ΔΔCt method and were normalized to the expression level of the house keeping gene GAPDH, which was used as an internal control.

## Immunohistochemistry

To check the amount and morphology of microglial cells in the brain tissue, brains were isolated and fixed in 4% paraformaldehyde (PFA) for 24 hr and then cryoprotected in 30% sucrose solution in phosphate buffered saline (PBS 1 x) for 24 hr. Samples were stored in Tissue-Tek O.C.T. compound (A. Hartenstein Laborversand) at –70 °C. 30 µm brains sections were cut using the Cryostat. Using the free floating method these slices were washed in 1 x PBS and blocked in 1 x PBS solution containing 0.2% Triton X-100, 10% goat serum and 1% BSA for 1 hr at room temperature (RT). Slices were incubated overnight at 4 °C with anti-ionized calcium-binding adaptor molecule 1 (IBA-1) (1:1000; rabbit polyclonal, Synaptic System) primary antibody diluted in 1 X PBS, 0.2% Triton X-100 and 10% goat serum. Cy3-conjugated AffiniPure Goat Anti-Rabbit IgG (H+L) (1:500; Jackson ImmunoResearch) was used as secondary antibody diluted in 1 X PBS. Sections were washed as before and stained with 4′,6-diamidino-2-phenylindole (DAPI) (SIGMA) followed by cover-slipping with Fluoro-gel with Tris buffer (Electron Microscopy Sciences).

## FACS analysis

Microglia activation was analyzed by measuring the marker CD68 with the FACS method (*Lonnemann et al., 2020*). Single cell isolation was performed using the Adult Brain Dissociation Kit (Miltenyi Biotec Order no. 130-107-677) from Miltenyi and the GentleMACS to homogenize the fresh isolated brains. Briefly, brain tissue was homogenized enzymatically and mechanistically for 30 min at 37 °C using the GentleMACS. In the following steps, the debris and the red blood cells were removed using the manufactory manual and products including a percoll based gradient and red blood cell lysis buffer. The cells were resuspended in FACS staining buffer (1xPBS +1% FCS+0.1% Na-Azide) and plated in V-bottom 96-well plate. Cells were stained for 30 min with CD11b-PerCP (1:50), CD45-APC (1:50), CD68-PE (1:50). The flowcytometry was measured using the BD LRS II SORP and analyzed with FlowJo Software.

## Methoxy-XO4 staining

The phagocytic activity regarding Aβ uptake in microglial cells of APP/PS1 and APP/PS1-IL37tg animals was analyzed by FACS. Here, WT, APP/PS1 and APP/PS1-IL37tg animals was used and crossed with the Cx3Cr1-GFP mouse line. The Cx3cr1-GFP transgenic mice express a microglia-specific (or monocytic-specific) green fluorescent protein in the CNS. To visualize Aβ particles animals were injected intraperitoneal with 10 mg/kg methoxy-XO4 (Abcam; ab142818; blood-brain barrier permeable amyloid-β fluorescent marker) in 50% DMSO/ 50% NaCl (0.9%) 3 hr prior to the start of experiments. After the brain was removed, the procedure was as in section 'FACS analysis'.

## Y-Maze test

There are several methods to evaluate cognitive function in rodents. Of these, one method is called 'Y-Maze' in which a mouse is placed in a maze with three equal arms (hence the term 'Y') each 32 cm from the center. First the mouse is free to explore the first two arms while the third arm is blocked. After the mouse get to know the existence of the two arms during 3 min training, the mouse was returned to its cage for 1.5 hr. Then the mouse is placed back to the Y-Maze with all three arms available. The observer records how often the mouse enters an arm. Due to animal's natural tendency to explore a newly introduced environment it is expected to be expressed by a high frequency of spontaneous alteration performance (SAP). The SAP score is a triplet of three successive different arm visits (ABC, BCA, CAB, BAC, ACB, CBA).

## Morris water maze test

The Morris water maze (MWM) test is an assay to analyze spatial memory formation and retention (*Morris, 1984*). A circular plastic pool (160 cm in diameter and 60 deep) filled up to 30 cm with opaque water (titanium dioxide, Euro OTC Pharma; water temperature 19–20°C) was used including a 10 cm escape platform submerged 1 cm below the water surface and three visual cues on the walls around the pool. Each day the test was performed in the same conditions (dim light and the same time of the day) by the same experimenter blind to all groups. The ANY-maze software (Stoelting, USA) with a camera above the center of the maze was used to track each trial. A pre-training with a visible platform was performed to guarantee the visual and swimming ability in all experimental groups and

in addition to get the animal used to the test situation. The pre-training lasted for three consecutive days with two trials each day (maximum of 60 s each) to reach the platform.

The mice were trained in the Morris water maze for 8 days with the invisible platform located in the northwest (NW) quadrant. Each day, animals were placed from randomly starting positions (SW, S, E and NE) for 4 trials in the water with 5-min intervals. The animals had a maximum of 60 s to find the platform otherwise they were guided to the platform and allowed to stay for additional 15 s. The memory retention was measured by performing a reference test. One reference test was performed on day 3 of the training acquisition (prior to the training session) and another reference test 24 hr after the last training day (day 9). The platform was removed during this reference test. The animals were tracked for 45 s (starting position SE).

## Electrophysiological experiments

To study learning and memory processes on cellular level, we did electrophysiological recording experiments in CA1 hippocampal neurons as described before (*Hosseini et al., 2018*; *Lonnemann et al., 2020*; *Hosseini et al., 2020*). Briefly, mice were deeply anesthetized with 100% $CO_2$, killed by decapitation following fast brain removal and transfer into ice-cold carbogenated (95% $O_2$ and 5% $CO_2$) artificial CSG (ACSF) containing 124 mM NaCl, 4.9 mM KCl, 1.2 mM $KH_2PO_4$, 2.0 mM $MgSO_4$, 2.0 mM $CaCl_2$, 24.6 mM $NaHCO_3$ and 10 D-glucose, pH 7.4. Acute hippocampal slices were prepared (400 µm) using a tissue chopper. The hippocampal slices were placed into an interface chamber (Scientific System Design) and incubated at 32 °C in a constant flow rate (0.5 ml/min) of carbogenated ACSF for 2 hr. Afterwards the recordings were started and field excitatory post synaptic potentials (fEPSPs) were measured in the stratum radiatum of the hippocampus CA1 sub-region. The Schaffer collateral pathway was stimulated using two monopolar, lacquer-coated stainless-steel electrodes (5 MΩ; AM Systems). Long-term potentiation (LTP) was induced after 20 min baseline recording by theta-burst stiulation (TBS) including four bursts at 100 Hz repeated 10 times in a 200ms interval. This stimulation was repeated three times in a 10 s interval. Only healthy sections with a stable baseline were included in the data set. Using the IntraCell software (version 1.5, LIN) the data set was analyzed.

## Golgi-Cox staining

To analyze the morphology of hippocampal neurons, Golgi-Cox staining was performed like previously described (*Hosseini et al., 2018*; *Lonnemann et al., 2020*; *Hosseini et al., 2020*). Briefly, mice were deeply anesthetized with $CO_2$ and sacrificed by decapitation. The brain was incubated in FD rapid Golgi-Cox staining kit according to the manufacturer's protocol. The tissue was incubated in a mixture of solution A (potassium dichromate and mercuric chloride) and B (potassium chromate) for at least 14 days at RT in the dark. After incubation of A-B the tissue was placed for 1 week in solution C (tissue protection) at RT. In the following, the brain was blocked in 2% agar and 200-µm-thick coronal sections were cut using a vibratome (Leica VT 1000 S). The slices were collected on gelatin-coated glass slides and stained with solution D and E before being dehydrated through graded alcohols and mounted using Permount (Thermo Fisher Scientific).

## Congo-Red staining

To examine the Aβ-plaques in APP/PS1 and APP/PS1-IL37 mice plaques were analyzed for the amount and size using the Congo-Red staining. Mice were deeply anesthetized with $CO_2$ and sacrificed via decapitation. Brains were isolated and hemispheres were fixed in 4% PFA for 24 hr and then cryoprotected in 30% sucrose solution in phosphate buffered saline (PBS 1 x). Hemispheres were stored in Tissue-Tek (Hartenstein Laborversand) at –70 °C and cutted using the Leica Cryostat (CM3050 S) in 30 µm slices. The sections were transported on gelatin-coated slides and in the following stained with the Congo-Red manufactorer's protocol using the FD Congo-Red Solution kit (FD Neurotechnologies, Inc) and mounted with Permount (Thermo Fisher Scientific).

## Imaging and image analysis

To analyze the hippocampal neuron morphology, CA1 and dentate gyrus (DG) cells were imaged in the three-dimensions (z-stack thickness of 0.5 µm) using Axioplan 2 imaging microscope (Zeiss) with a 63 x (N.A. 1) oil objective equipped with a digital camera (AxioCam MRm, Zeiss). All selected dendrites were analyzed per spine density via number of spines (counted manually using the ImageJ

software) per micrometer of dendritic length more than 50–60 µm which were positioned at least 40–50 µm away from the cell soma.

To analyze microglia morphology microscopic images of anti-IBA-1 were taken within the area of cortex and hippocampus. Images were taken in 3D (z-stack thickness 1 µm) using Axioplan 2 imaging microscope (Zeiss) equipped with an ApoTome module (Zeiss) with a 20 X objective (NA, 0.8) and digital camera (AxioCam MRm; Zeiss). IBA-1 positive cells were counted with clearly visible nuclei by DAPI staining for microglia density and the processes of IBA-1 positive cells were analyzed to investigate the activation status of these cells by using the ImageJ software (Wane Rasband NIH, USA).

To survey the Aβ-plaques images from Congo-Red stained slices were taken. Congo-Red staining presents a bright fluorescence emission at 614 nm with excitation at 497 nm. Images of brain sections were taken using an Axioplan 2 imaging microscope (Zeiss) with a 2.5 X objective (N.A. 0.07) connected to a digital camera (Nikon) with the same light exposure time of 1 s in all groups. Plaque load and plaque size were analyzed using the ImageJ software (Wane Rasband NIH, USA) with the analyze particle tool. The polygon selection tool was used to generate the region of interest (ROI) for the area of hippocampus and cortex. Plaque load (number of particles) and plaque size (area of particles) were normalized to the area of hippocampus and cortex and plotted as plaque load and plaque size per mm$^2$.

## Extraction of intracellular Metabolites for Gas Chromatography-Mass Spectrometry (GC-MS)

Primary microglia cells ($5*10^5$ cells) were plated onto 12-well plates and incubated for 24 hr. The medium was exchanged with fresh medium or medium mixed with 10 ng/ml LPS for 24 hr. To extract intracellular metabolites cells were washed once with 0.9% NaCl and 500 µL of a cold methanol/water mixture was added. The water fraction contained the internal standard (IS) D6-glutaric-acid (c=1 µg/ml). Cells were scraped and transferred into 250 µL of –20 °C Chloroform. After vortexing samples for 20 min at 4 °C with maximal speed in an automatic shaker, samples were centrifuge for 5 min at above 15,000 xg in a table centrifuge (4 °C). A total of 200 µL of the upper phase was transferred in a GC glass vial with micro insert and evaporated to dry under vacuum at 4 °C, overnight. To avoid condensation, the GC glass vials were warmed to room temperature under vacuum and capped afterwards with magnetic caps.

## Metabolite measurement

Derivatization of the samples was performed directly before GC-MS measurement. Metabolite extracts were dissolved in 15 µL pyridine, containing 20 mg/mL methoxyamine hydrochloride at 55 °C for 90 min under shaking. After adding 15 µL N-methyl-N-tert-butyldimethylsilyl-trifluoroacetamide samples were incubated at 55 °C for 60 min under continuous shaking. GC/MS analysis was performed using an Agilent 7890B GC coupled to an Agilent 5977B MSD. The gas chromatogram was equipped with a 30 m ZB-35 Phenomenex 5 m Guard capillary column. As carrier gas helium was used at a flow rate of 1.0 ml/min. A sample volume of 1 µL was injected into a split/splitless inlet, operating a splitless mode at 270 °C. The GC oven Temperature was held at 100 °C for 2 min and increased to 300 at 10 °C/min and held for further 4 min. Afterwards the temperature was increased to 325 °C. The MSD was operating under electron ionization at 70 eV. The MS source was held at 230 °C and the quadrupole at 150 °C. The total run time of one sample was 26 min. Full scan mass spectra were acquired from m/z 70 to m/z 800. All GC-MS chromatograms were processed using Metabolite Detector, v3.020151231Ra (*Hiller et al., 2009*).

## RNA sequencing

Briefly, brain tissue from WT and IL –37tg mice treated with either saline or LPS was homogenized enzymatically and mechanically for 30 min at 37 °C using GentleMACS. Cellular debris and red blood cells were removed using the manufacturer's manual and products such as a Percoll-based gradient and a red blood cell lysis buffer. Microglia were then isolated by administration of CD11b MicroBeads followed by magnetic bead separation. RNA was then isolated using the Nucleo Spin RNA Isolation Kit (Machery-Nagel). The cDNA synthesis was performed and libraries were prepared using the NEBNext Ultra II Directional RNA Library Prep Kit for Illumina at the Genome Analytics of the Helmholtz Center for Infection Research. The libraries were sequenced on Illumina NovaSeq 6000 using

**Table 1.** Statistical table.

| Figure | Name | Test | | Multi-comparison | | |
|---|---|---|---|---|---|---|
| 1B | 100 ng/ml 6 hr | One-way ANOVA | $F_{(2,9)}=2,428$; p=0.1434 | Tukey's | 0,8947 | WT vs. HET |
| | | | | | 0,1460 | WT vs. HOM |
| | 100 ng/ml 24 hr | One-way ANOVA | $F_{(2,11)}=7,941$; p=0.0073 | Tukey's | 0,0539 | WT vs. HET |
| | | | | | 0,0067 | WT vs. HOM |
| | 1 µg/ml 6 hr | One-way ANOVA | $F_{(2,9)}=12,74$; p=0.0024 | Tukey's | 0,0090 | WT vs. HET |
| | | | | | 0,0037 | WT vs. HOM |
| | 1 µg/ml 24 hr | One-way ANOVA | $F_{(2,11)}=3,915$; p=0.0520 | Tukey's | 0,3586 | WT vs. HET |
| | | | | | 0,0426 | WT vs. HOM |
| 1C | IL-6 10 ng/ml | ttest | t=1,742 df = 16; p=0.1007 | | | WT vs. IL37tg |
| | IL-6 100 ng/ml | ttest | t=3,597 df = 16; p=0.0024 | | | WT vs. IL37tg |
| | IL-6 1 µg/ml | ttest | t=5,127 df = 16; p=0.0001 | | | WT vs. IL37tg |
| 1D | TNF-α 10 ng/ml | ttest | t=3,376 df = 16; p=0.0038 | | | WT vs. IL37tg |
| | TNF-α 100 ng/ml | ttest | t=6,774 df = 16; p<0.0001 | | | WT vs. IL37tg |
| | TNF-α 1 µg/ml | ttest | t=6,264 df = 16; p<0.0001 | | | WT vs. IL37tg |
| 1E | IL-1β 10 ng/ml | ttest | t=2,390 df = 16; p=0.0295 | | | WT vs. IL37tg |
| | IL-1β 100 ng/ml | ttest | t=5,027 df = 16; p=0.0001 | | | WT vs. IL37tg |
| | IL-1β 1 µg/ml | ttest | t=5,852 df = 16; p<0.0001 | | | WT vs. IL37tg |
| 1F | IL-37 mRNA | One-way ANOVA | $F_{(7,15)}=2,629$; p=0.0550 | Fisher's LSD | 0.0171 | 0 hr vs. 1 hr |
| | | | | | 0.0086 | 0 hr vs. 4 hr |
| | | | | | 0.0238 | 0 hr vs. 24 hr |
| 1G | IL-6 | ttest | t=20,39 df = 4; p<0.0001 | | | WT vs. IL37tg |
| 1H | rIL-37 100 ng/ml | One-way ANOVA | $F_{(2,30)}=5,135$; p=0.0121 | Tukey's | 0.1675 | WT vs. 100 ng |
| | | | | | 0.0133 | WT vs. 500 ng |
| | rIL-37 1 µg/ml | One-way ANOVA | $F_{(2,30)}=3,512$; p=0.0426 | Tukey's | 0.2457 | WT vs. 100 ng |
| | | | | | 0.0498 | WT vs. 500 ng |
| 2B | Itaconate Treatment | Two-way ANOVA | $F_{(1,20)}=174,9$; p<0.0001 | Bonferroni's | <0.0001 | WT NIL vs. LPS |
| | | | | | <0.0001 | IL37 NIL vs. LPS |
| | Itaconate Genotype | Two-way ANOVA | $F_{(1,20)}=27,55$; p<0.0001 | Bonferroni's | >0.9999 | NIL WT vs. IL37 |
| | | | | | <0.0001 | LPS WT vs. IL37 |
| 2C | Succinate Treatment | Two-way ANOVA | $F_{(1,20)}=98,40$; p<0.0001 | Bonferroni's | <0.0001 | WT NIL vs. LPS |
| | | | | | 0.0005 | IL37 NIL vs. LPS |
| | Succinate Genotype | Two-way ANOVA | $F_{(1,20)}=17,01$; p=0.0005 | Bonferroni's | >0.9999 | NIL WT vs. IL37 |
| | | | | | <0.0001 | LPS WT vs. IL37 |
| 3B | Body weight Treatment | Two-way ANOVA | $F_{(1,17)}=220,5$; p<0.0001 | Bonferroni's | <0.0001 | WT NIL vs. LPS |

*Table 1 continued on next page*

*Table 1 continued*

| Figure | Name | Test | | Multi-comparison | | |
|---|---|---|---|---|---|---|
| | | | | | <0.0001 | IL37 NIL vs. LPS |
| | Body weight Genotype | Two-way ANOVA | F(1,41)=1,486; p=0.2299 | Bonferroni's | >0.9999 | NIL WT vs. IL37 |
| | | | | | 0.0139 | LPS WT vs. IL37 |
| 3D | CD68 Treatment | Two-way ANOVA | F(1,6)=19,52; p=0.0045 | Bonferroni's | 0.0017 | WT NIL vs. LPS |
| | | | | | >0.9999 | IL37 NIL vs. LPS |
| | CD68 Genotype | Two-way ANOVA | F(1,6)=48,42; p=0.0004 | Bonferroni's | 0.0167 | NIL WT vs. IL37 |
| | | | | | <0.0001 | LPS WT vs. IL37 |
| 3E | IL-1β Treatment | Two-way ANOVA | F(1,7)=9,399;p=0.0182 | Bonferroni's | 0.0244 | WT NIL vs. LPS |
| | | | | | 0.7541 | IL37 NIL vs. LPS |
| | IL-1β Genotype | Two-way ANOVA | F(1,7)=3,182;p=0.0935 | Bonferroni's | >0.9999 | NIL WT vs. IL37 |
| | | | | | 0.0366 | LPS WT vs. IL37 |
| 3F | Iba1 CA1 Treatment | Two-way ANOVA | F(1,25)=5,222;p=0.0311 | Bonferroni's | 0.0024 | WT NIL vs. LPS |
| | | | | | >0.9999 | IL37 NIL vs. LPS |
| | Iba1 CA1 Genotype | Two-way ANOVA | F(1,34)=4,951;p=0.0328 | Bonferroni's | >0.9999 | NIL WT vs. IL37 |
| | | | | | 0.0027 | LPS WT vs. IL37 |
| 3G | Processes CA1 Treatment | Two-way ANOVA | F(1,46)=1,066; p=0.3073 | Bonferroni's | >0.9999 | WT NIL vs. LPS |
| | | | | | 0.6229 | IL37 NIL vs. LPS |
| | Processes CA1 Genotype | Two-way ANOVA | F(1,58)=1,238; p=0.2704 | Bonferroni's | 0.5670 | NIL WT vs. IL37 |
| | | | | | >0.9999 | LPS WT vs. IL37 |
| 3H | Iba1 Cortex Treatment | Two-way ANOVA | F(1,25)=15,97; p=0.0005 | Bonferroni's | 0.0002 | WT NIL vs. LPS |
| | | | | | 0.3234 | IL37 NIL vs. LPS |
| | Iba1 Cortex Genotype | Two-way ANOVA | F(1,34)=5,159; p=0.0296 | Bonferroni's | >0.9999 | NIL WT vs. IL37 |
| | | | | | 0.0054 | LPS WT vs. IL37 |
| 3I | Processes Cortex Treatment | Two-way ANOVA | F(1,104)=9,865; p=0.0022 | Bonferroni's | 0.1401 | WT NIL vs. LPS |
| | | | | | 0.0232 | IL37 NIL vs. LPS |
| | Processes Cortex Genotype | Two-way ANOVA | F(1,104)=0,119; p=0.7308 | Bonferroni's | >0.9999 | NIL WT vs. IL37 |
| | | | | | 0.8147 | LPS WT vs. IL37 |
| 4A | LTP WT | Two-way ANOVA | F(1,28)=4,459; p=0.0438 | Fisher's LSD Time point 43–80 | <0.05 | WT NIL vs. LPS |
| 4B | LTP IL37 | Two-way ANOVA | F(1,23)=0,085; p=0.7734 | | | IL37 NIL vs. LPS |
| 4C | LTP last 5 min Treatment | Two-way ANOVA | F(1,22)=7,887;p=0.0102 | Bonferroni's | 0.0043 | WT NIL vs. LPS |
| | | | | | >0.9999 | IL37 NIL vs. LPS |

*Table 1 continued on next page*

*Table 1 continued*

| Figure | Name | Test | | Multi-comparison | | | |
|--------|------|------|---|------------------|---|---|---|
| | LTP last 5 min Genotype | Two-way ANOVA | F(1,29)=1,552; p=0.2228 | | | | |
| 4D | Spines CA1 Treatment | Two-way ANOVA | F(1,32)=22,81; p<0.0001 | Bonferroni's | <0.0001 | | WT NIL vs. LPS |
| | | | | | 0.1145 | | IL37 NIL vs. LPS |
| | Spines CA1 Genotype | Two-way ANOVA | F(1,44)=26,68; p<0.0001 | Bonferroni's | 0.0657 | | NIL WT vs. IL37 |
| | | | | | <0.0001 | | LPS WT vs. IL37 |
| 4E | Spines DG Treatment | Two-way ANOVA | F(1,26)=3,307; p=0.0805 | Bonferroni's | 0.0062 | | WT NIL vs. LPS |
| | | | | | >0.9999 | | IL37 NIL vs. LPS |
| | Spines DG Genotype | Two-way ANOVA | F(1,36)=0,529; p=0.4718 | Bonferroni's | 0.5222 | | NIL WT vs. IL37 |
| | | | | | 0.0295 | | LPS WT vs. IL37 |
| 4G | LTP WT | Two-way ANOVA | F(1,30)=4,925; p=0.0342 | Fisher's LSD Time point 39–80 | <0.05 | | WT NIL vs. LPS |
| 4H | LTP IL37 | Two-way ANOVA | F(1,26)=0,01395; p=0.9069 | | | | IL37 NIL vs. LPS |
| 4I | LTP last 5 min Treatment | Two-way ANOVA | F(1,27)=4,613; p=0.0409 | Bonferroni's | 0.0285 | | WT NIL vs. LPS |
| | | | | | >0.9999 | | IL37 NIL vs. LPS |
| | LTP last 5 min Genotype | Two-way ANOVA | F(1,29)=1,747; p=0.1966 | | | | |
| 5A | Y-Maze Treatment | Two-way ANOVA | F(1,6)=18,25; p=0.0052 | Bonferroni's | 0.0046 | | WT NIL vs. IL1 |
| | | | | | 0.7313 | | IL37 NIL vs. IL1 |
| | Y-Maze pre-treatment | Two-way ANOVA | F(1,6)=9,899; p=0.0199 | Bonferroni's | >0.9999 | | NIL WT vs. IL37 |
| | | | | | 0.0024 | | IL1 WT vs. IL37 |
| 5B | IL-6 Treatment | Two-way ANOVA | F(1,16)=264,6; p<0.0001 | Bonferroni's | <0.0001 | | WT NIL vs. IL1 |
| | | | | | <0.0001 | | IL37 NIL vs. IL1 |
| | IL-6 pre-treatment | Two-way ANOVA | F(1,16)=10,27; p=0.0055 | Bonferroni's | 0.8327 | | NIL WT vs. IL37 |
| | | | | | 0.0004 | | IL1 WT vs. IL37 |
| 5C | IL-1α Treatment | Two-way ANOVA | F(1,16)=132,0; p<0.0001 | Bonferroni's | <0.0001 | | WT NIL vs. IL1 |
| | | | | | <0.0001 | | IL37 NIL vs. IL1 |
| | IL-1α pre-treatment | Two-way ANOVA | F(1,16)=17,47; p=0.0007 | Bonferroni's | >0.9999 | | NIL WT vs. IL37 |
| | | | | | 0.0003 | | IL1 WT vs. IL37 |
| 6B | IL-1β | One-way ANOVA | F(2,19)=5,224; p=0.0156 | Fisher's LSD | 0.0903 | | WT vs. APP |
| | | | | | 0.2480 | | WT vs. APP-IL37 |
| | | | | | 0.0046 | | APP vs. APP-IL37 |
| 6C | IL-6 | One-way ANOVA | F(2,21)=3,600; p=0.0452 | Fisher's LSD | 0.0220 | | WT vs. APP |
| | | | | | 0.5264 | | WT vs. APP-IL37 |
| | | | | | 0.0466 | | APP vs. APP-IL37 |
| 6E | CD68 9–12 m | One-way ANOVA | F(2,11)=14,39; p=0.0008 | Bonferroni's | 0.0008 | | WT vs. APP |

*Table 1 continued on next page*

*Table 1 continued*

| Figure | Name | Test | | Multi-comparison | | |
|---|---|---|---|---|---|---|
| | | | | | 0.0529 | WT vs. APP-IL37 |
| | | | | | 0.04494 | APP vs. APP-IL37 |
| 6F | CD68 6 m | One-way ANOVA | F(2,12)=13,45; p=0.0009 | Bonferroni's | 0.0007 | WT vs. APP |
| | | | | | 0.0263 | WT vs. APP-IL37 |
| | | | | | 0.0985 | APP vs. APP-IL37 |
| 6G | CD68 20–23 m | One-way ANOVA | F(2,12)=13,38; p=0.0009 | Bonferroni's | 0.0007 | WT vs. APP |
| | | | | | 0.0359 | WT vs. APP-IL37 |
| | | | | | 0.0383 | APP vs. APP-IL37 |
| 6I | Plaque load Hp | ttest | t=2,403 df = 74; p=0.0187 | | | APP vs. APP-IL37 |
| | Plaque load Cx | ttest | t=4,953 df = 63; p<0.0001 | | | APP vs. APP-IL37 |
| 6J | Plaque size Hp | ttest | t=2,296 df = 74; p=0.0245 | | | APP vs. APP-IL37 |
| | Plaque size Cx | ttest | t=5,949 df = 63; p<0.0001 | | | APP vs. APP-IL37 |
| 6G | Abeta uptake | One-way ANOVA | F(2,22)=136; p<0.0001 | Bonferroni's | 0.0001 | WT vs. APP |
| | | | | | 0.0001 | WT vs. APP-IL37 |
| | | | | | 0.0026 | APP vs. APP-IL37 |
| 7B | Latency WT | One-way ANOVA | F(7,88)=5,842; p<0.0001 | | | WT |
| | Latency APP | One-way ANOVA | F(7,72)=3,477; p=0.0029 | | | APP |
| | Latency APP-IL37 | One-way ANOVA | F(7,80)=4,445; p=0.0003 | | | APP-IL37 |
| | Latency | 2-way ANOVA | F(2,30)=10,50; p=0.0003 | Fisher's LSD Day 1 | <0.05 | WT vs. APP |
| | | | | Day 3–6 | <0.01 | |
| 7C | PT WT | ttest | t=11,45 df = 22; p<0,0001 | | | NT vs. TQ |
| | PT APP | ttest | t=0,9874 df = 18; p=0.3365 | | | NT vs. TQ |
| | PT APP-IL37 | ttest | t=3,705 df = 20; p=0.0014 | | | NT vs. TQ |
| | PT TQs | One-way ANOVA | F(2,30)=8,415; p=0.0013 | Turkey's | 0.0009 | WT vs. APP |
| | | | | | 0,0505 | WT vs. APP-IL37 |
| 7E | LTP | Two-way ANOVA | F(2,45)=9,286; p=0.0004 | Fisher's LSD Time point 21–80 | <0.0005 | WT vs. APP |
| | | | | Fisher's LSD Time point 41–80 | <0.05 | WT vs. APP-IL37 |
| 7F | Mean LTP 20–25 min | One-way ANOVA | F(2,45)=7,090; p=0.0021 | Turkey's | 0.0016 | WT vs. APP |
| | | | | | 0.5444 | WT vs. APP-IL37 |
| 7G | Mean LTP 75–80 min | One-way ANOVA | F(2,45)=9,354; p=0.0004 | Turkey's | 0.0014 | WT vs. APP |
| | | | | | 0.0023 | WT vs. APP-IL37 |
| 7I | Spine density | One-way ANOVA | F(2,63)=6.318;p=0.0032 | Turkey's | 0.0026 | WT vs. APP |
| | | | | | 0.0536 | WT vs. APP-IL37 |

NovaSeq 6000 S1 Reagent Kit (100 cycles, paired end run) with an average of 5x107 reads per RNA sample. Before alignment to reference genome (mm10) each sequence in the raw FASTQ files were trimmed on base call quality and sequencing adapter contamination using Trim Galore! wrapper tool. Reads shorter than 20 bp were removed from FASTQ file (*Andrews, 2010*; *Krueger, 2012*). Trimmed reads were aligned to the reference genome using open source short read aligner STAR (https://code.google.com/p/rna-star/) with settings according to log file (*Dobin et al., 2013*). Feature counts were determined using R package 'Rsubread' (*Liao et al., 2014*; *Durinck et al., 2005*). Only genes showing counts greater 5 at least two times across all samples were considered for further analysis (data cleansing). Gene annotation was done by R package 'bioMaRt' Before starting the statistical analysis steps, expression data was $\log_2$ transform and TMM normalized (edgeR). Differential gene expression was calculated by R package 'edgeR'. Functional analysis was performed by R package 'clusterProfiler' (*Robinson et al., 2010*; *Yu et al., 2012*).

### Statistical analysis

Sample size for all experiments were calculated a priory with G*Power software (HHU Düsseldorf). Samples were allocated into the experimental groups randomly.

Data were analyzed and plotted by GraphPad Prism 8 (GraphPad Software, Inc USA) and presented as mean ± SEM. Statistically analysis were performed with either unpaired t-test, one-way ANOVA (post hoc test Fisher's LSD or Turkey's multiple comparisons) or two-way ANOVA (post hoc test Fisher's LSD or Bonferroni's multiple comparisons) depending on experiments. The minimum significance value was considered as $p < 0.05$ (*Table 1*).

## Acknowledgements

This work was in part supported by the DFG (SFB854), and by the Helmholtz-Gemeinschaft, Zukunftsthema "Immunology and Inflammation" (ZT-0027) to MK, and NIH Grant AI-15614 (to CAD). We acknowledge Philip Bufler, for providing us the hIL-37tg mouse and Andreas Holz for excellent guidance and critical comments to an earlier version of the manuscript.

## Additional information

### Funding

| Funder | Grant reference number | Author |
|---|---|---|
| Deutsche Forschungsgemeinschaft | SFB854 | Martin Korte |

The funders had no role in study design, data collection and interpretation, or the decision to submit the work for publication.

### Author contributions

Niklas Lonnemann, Data curation, Formal analysis, Visualization, Methodology, Writing - original draft; Shirin Hosseini, Formal analysis, Writing - original draft; Melanie Ohm, Formal analysis, Visualization, Methodology; Robert Geffers, Formal analysis, Methodology; Karsten Hiller, Resources, Supervision, Methodology; Charles A Dinarello, Conceptualization, Writing – review and editing; Martin Korte, Conceptualization, Data curation, Software, Supervision, Funding acquisition, Writing - original draft, Project administration, Writing – review and editing

### Author ORCIDs

Niklas Lonnemann (iD) http://orcid.org/0000-0002-7285-9995
Shirin Hosseini (iD) http://orcid.org/0000-0001-7949-862X
Martin Korte (iD) http://orcid.org/0000-0001-6956-5913

### Ethics

All experimental procedures and protocolls were authorized by the animal welfare representative of the TU Braunschweig and the LAVES of the state of Lower Saxony in Germany (Oldenburg, Germany) (33.19-42502-04-16/2170).

Decision letter and Author response
Decision letter https://doi.org/10.7554/eLife.75889.sa1
Author response https://doi.org/10.7554/eLife.75889.sa2

## Additional files

### Supplementary files
• Transparent reporting form

### Data availability
All data generated or analysed during this study are included in the manuscript and supporting file; Source Data files have been provided for all Figures.

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
