## [Editor Report]

In this manuscript, the authors demonstrated that acute and chronic neuroinflammation was attenuated in human IL-37 (hIL-37) transgenic mice, thus revealing the effects of an anti-inflammatory cytokine hIL-37 in the central nervous system of mice. This study will be of interest to scientists studying neuroinflammation and searching for potential therapeutic targets.

---

## [Decision Letter]

**Decision letter after peer review:**

Thank you for submitting your article "IL-37 expression reduces acute and chronic neuroinflammation and rescues cognitive impairment in an Alzheimer's disease mouse model" for consideration by *eLife*. Your article has been reviewed by 3 peer reviewers, one of whom is a member of our Board of Reviewing Editors, and the evaluation has been overseen by Jeannie Chin as the Senior Editor. The reviewers have opted to remain anonymous.

Essential revisions:

1. The expression profile of hIL-37 receptors on different cell types under different conditions should be shown. The phenotype seen in microglia in vitro was attributed to hIL-37 acting on microglia themselves, but it is not clear this is the only possibility.

2. Experiments using human cell lines or patient samples would provide additional validation.

3. The authors mentioned that in acute inflammation, little LPS is likely to enter the brain because of an intact blood-brain barrier (BBB). It is also possible that the peripherally-induced inflammation may contribute to the neuroinflammation centrally. Due to the systemic overexpression of hIL-37, to exclude the peripheral effects of hIL-37, the experiments of direct injection of hIL-37 into the brain would help clarify the conclusion.

4. Background, Lines 87,88: The overall premise that anti-inflammatory therapies will be beneficial in AD is overly simplistic. The inflammatory activation of glial cells is required as a protective immune response, including pro-inflammatory activation. Also, retrospective studies that have demonstrated that NSAIDs are not a successful therapeutic strategy go against this notion. Please consider rephrasing.

5. What is known about IL-37 expression in humans? Since it is expressed by all cells in the current model, will this hamper biological interpretation, since the brain RNAseq database suggests that it is highly enriched specifically in microglia? This confounding factor makes biological interpretation of the in-vivo data a little difficult.

6. Results, Page 4, Lines 148-152: Description of Figure 1C-1E is confusing and unclear as written. Please consider rephrasing.

7. Results, Page 5: The points about the TCA cycle metabolites is raised in this section for the first time. Please consider introducing this concept in relation to AD / related immune responses in the introduction.

8. Figure 3C: Please include what the X-axis is plotting? As a supplemental figure, please include your initial gating strategy used for all flow cytometry data. Are there any CD68+CD11b+ CD45 high cells in these animals? If an effect of IL-37 is seen in filtration of peripheral inflammatory macrophages into the brain, this is a very strong data set to improve the impact of this figure.

9. Results, page 7: The authors make the statement that IL-37 signaling has a significant impact on the pro-inflammatory activation of microglia in the brain. However, this is based on just the measurement of IL-1b and microglial numbers. The authors should measure other markers of inflammation that are known to go up in the brain upon LPS challenge. Maybe using an Elisa based multiplex assay such as MSD or Luminex to measure brain inflammatory cytokines will add to this data set.

10. Results, page 7: Figure 4A-4B: Was the LPS injection used for these experiments the same as outlined for figure 3? Please clarify this in the method section, especially since 2 different LPS treatment paradigms have been used for 4A-4B and 4G-4H.

11. The paradigm for the Y-maze testing (pre-treatment with IL-37, followed by IL1b), though in-line with the current study, is not biologically relevant. Since IL-37 will be upregulated as part of the neuroinflammatory response, have the authors tried to treat animals with IL-37 after IL-1b or concurrently with IL-1B?

12. Figure 1B: This figure suggests that 1 i.p injection with rIL-1B completely abolishes any working memory In WT mice. This is surprising. Please clarify.

13. For Figures6 and 7: Were both sexes used for APPPS1 studies? If so, please clarify in the methods section as to the number of males and females, given that female APPPS1 mice show more aggressive disease as compared to males.

14. Figure 6. Please show all gating strategies as well as relevant gates drawn to identify CD11b+ CD45low microglia, as well as the methoxy X-04+ vs methoxy X-04- microglia. Is there a CD11b+ CD45high microglial subset at this age, since at earlier timepoints, these distinct microglial subsets are evident and have different activation profiles.

15. Please include details of how brain tissue was processed to prepare single-cell suspension for all FACS data. i.e: percoll based density gradients or myelin removal beads etc?

16. Authors discuss their methoxy-X04 data as indicative of "increased clearance" of plaques. Careful interpretation of this data set is required as this assay only measures Abeta "uptake". Since no measures of clearance have been assessed, please reword and/ reinterpret this in the results and Discussion section.

17. Please include representative images for plaque loads in the cortex that correlate with quantification in 6I. Also, please include a representative image for differences in plaque size.

18. Figure 7A: Please mention the average age of the animals used in the 3 animal groups for MW testing. 9-12 months is a big age range for these animals and results will be skewed if one cohort is largely composed of older animals vs. younger.

19. The Discussion section is rather long and largely used to re-summarize results. The authors should reconsider using this section to put their data into context for the big-picture field of neurodegeneration. Some suggestions – How do the authors picture IL-37 signaling to play a role in human AD?

a. Given the detailed description of distinct microglial subsets associated with amyloid vs. tau pathology in AD, do the authors anticipate IL-37 to have distinct effects depending on pathological outcomes?

b. Literature has demonstrated that IL-1b, IL-6 and TNFa from reactive microglia may play a role in exacerbating tau-spread. The authors should discuss the implications of their findings in this light

c. Is anything is known about IL-37 signaling in other neuroinflammatory models (injury, MS etc)? A possible discussion of these in the context of neuroprotection may strengthen this section.

20. in vitro phenotypes in Figure 1 were not closely connected with the following in vivo neuroinflammation phenotypes. Although three different neuroinflammation models were used, the detailed mechanisms for how IL-37 reduced neuroinflammation were not well addressed.

21) Figure 3 and Figure 5 showed reduced pro-inflammatory cytokine production in brain lysates. However, glial cells other than microglia, such as astrocytes and infiltrating leukocytes could be sources of inflammatory cytokines. There was no evidence supporting that reduced cytokine levels were intrinsic to microglia as modelled in the in vitro system. Thus, in vitro and in vivo phenotypes appeared somewhat disconnected.

22) In Figure 5, IL-1β-induced neuroinflammation model was used to demonstrate the beneficial effects of IL-37 on cognition and synaptic function. Could injection of recombinant IL-37 display similar effects for LPS challenge model in Figure 3 and Figure 4?

23) Reduced IL-37-mediated neuroinflammation was shown in both chronic and acute models in the manuscript. Did microglia or other cells play a major role in those models? The detailed mechanisms regarding cell type contribution were largely missing in the manuscript.

24. Figure 2 showed differential metabolomic profiling of microglial cells between WT and IL-37tg mice. However, there was no further evidence demonstrating that the metabolic function of microglial cells was indeed altered by IL-37 expression. It would be better to show the results of seahorse or other metabolic functional assays.

25. The models for acute neuroinflammation including LPS and IL-1β challenge were systematic inflammation. It might be reasonable to propose that reduced neuroinflammation was a secondary effect to reduced inflammation response in the periphery. In addition, in Figure 3, would injection of LPS twice induce tolerance responses?

26. In Figure 3 and Figure 6, CD68 was used as an activation marker for microglia. However, CD68 expression by itself is not enough to define microglia to be in the activation state. The phenotypic changes of microglial cells would also depend on specific models used. Additional experimental evidence is needed for defining the reduced activation status of microglia in IL-37tg mice.

---

## [Author Response]

Essential revisions:1. The expression profile of hIL-37 receptors on different cell types under different conditions should be shown. The phenotype seen in microglia in vitro was attributed to hIL-37 acting on microglia themselves, but it is not clear this is the only possibility.

It has been shown that IL-37 requires the receptors IL-18Rα and IL-1R8 (SIGIRR) to exert its versatile anti-inflammatory effects in innate signal transduction. Based on the previous studies now listed in the revised manuscript, astrocytes are also capable of expressing the required receptors for IL-37. However, in the new experiments we performed, we stimulated primary astrocytes with LPS and then measured the levels of IL-6 and TNF-α (see Sup. Figure 1). In contrast to primary microglia from IL-37tg mice (Figure 1C-D), levels of pro-inflammatory cytokines were not reduced in primary astrocytes from IL-37tg mice after 100ng/ml LPS stimulation for 24 hours (see Sup. Figure 1A-B). Therefore, we can assume that although astrocytes also express receptors for IL-37, IL-37 signalling may not occur via them because the anti-inflammatory consequences are not expressed. These new experiments are added in the supplementary section and discussed in the revised manuscript.

A new supplementary Figure 1 is added. The changes can be found in lines 118-124 and 171-175, respectively.

2. Experiments using human cell lines or patient samples would provide additional validation.

Although the suggestion to use human material could significantly strengthen the results of this manuscript, performing new experiments with human cell lines requires a large time window because our laboratory currently has no experience with this. In addition, obtaining patient samples requires official ethical approval, which takes a lot of time. Therefore, in the revised manuscript, in the Introduction and Discussion, we try to refer to the recent human studies on the role of IL-37 in various human diseases such as autism and multiple sclerosis.

The changes can be found in lines 120-124 and 731-734, respectively.

3. The authors mentioned that in acute inflammation, little LPS is likely to enter the brain because of an intact blood-brain barrier (BBB). It is also possible that the peripherally-induced inflammation may contribute to the neuroinflammation centrally. Due to the systemic overexpression of hIL-37, to exclude the peripheral effects of hIL-37, the experiments of direct injection of hIL-37 into the brain would help clarify the conclusion.

In this study, we used IL-37tg mice that received i.p. injection of LPS to investigate the potential anti-inflammatory effects of IL-37 on the acute deleterious inflammatory response induced by LPS in the brain. We agree with the comment that containment of the inflammatory response in the periphery by IL-37 could prevent the inflammatory consequences in the brain, and we were not sure whether local expression of IL-37 in the brain could be beneficial in blocking the effects of LPS. Therefore, in a separate ex vivo approach, we treated the acute hippocampal slices of IL-37tg mice with LPS and then measured long-term potentiation (LTP) using electrophysiological experiments. Although LTP was impaired when acute slices from WT mice were treated with LPS, this was not the case for slices from IL-37tg mice. Subsequent mRNA expression in acute slices showed that IL-37 expression was higher in slices from IL-37tg mice after LPS stimulation (Data not shown because n was low due to the small amount of materials). Thus, we can conclude that in addition to peripheral IL-37, local induction in the brain during LPS administration may be part of the rescue mechanisms. In our revised manuscript, we have made this point more explicit.

The information can be found in lines 291-308.

4. Background, Lines 87,88: The overall premise that anti-inflammatory therapies will be beneficial in AD is overly simplistic. The inflammatory activation of glial cells is required as a protective immune response, including pro-inflammatory activation. Also, retrospective studies that have demonstrated that NSAIDs are not a successful therapeutic strategy go against this notion. Please consider rephrasing.

As the editor mentioned, although acute inflammation is part of the body's response to tissue damage and is critical to the healing process, a prolonged inflammatory response can be harmful. Chronic inflammation has been shown to affect and accelerate each of the three hallmarks of AD pathology, including Aβ accumulation, tau phosphorylation, and cognitive decline associated with loss of synapses and neurons. Therefore, attenuating chronic inflammatory processes may be a useful way to treat or prevent the progression of AD. NSAIDs are generally used for their anti-inflammatory, analgesic, and antipyretic effects. NSAIDs work by blocking the production of prostaglandins by inhibiting two cyclooxygenase enzymes. However, prostaglandins contribute to the development of the major signs of acute inflammation. Therefore, as the editor mentioned, they are not a useful therapy to prevent the progression of AD as a result of a chronic inflammatory response.

We have taken this point into consideration and reworded the mentioned part in the introduction to make it clearer.

The information can be found in lines 89-91.

5. What is known about IL-37 expression in humans? Since it is expressed by all cells in the current model, will this hamper biological interpretation, since the brain RNAseq database suggests that it is highly enriched specifically in microglia? This confounding factor makes biological interpretation of the in-vivo data a little difficult.

Previously, IL-37 was shown to be constitutively expressed in various human tissues and cells, which may help in maintaining immune homeostasis. IL-37 is expressed in human immune cells mainly in circulating monocytes, tissue macrophages, dendritic cells, tonsil B cells, and plasma cells. In addition, IL-1β and TNF-α were shown to increase the expression of IL-37 in cultured human microglia. Review of the expression profile of human iPSC derived microglia after LPS treatment using the GEO database also reveals the presence of IL-37

(https://www.ncbi.nlm.nih.gov/geo/query/acc.cgi?acc=GSE186301), but it is still not clear which cell types in the human brain produce IL-37 abundantly. In the revised manuscript, the previously known information about IL-37 in humans was added in the Introduction and Discussion sections.

The information can be found in lines 123-127 and 550-554.

6. Results, Page 4, Lines 148-152: Description of Figure 1C-1E is confusing and unclear as written. Please consider rephrasing.

We followed the editor’s advice and rephrased this part.

7. Results, Page 5: The points about the TCA cycle metabolites is raised in this section for the first time. Please consider introducing this concept in relation to AD / related immune responses in the introduction.

The relevant information was added in the revised manuscripts. The information can be found in lines 131-136.

8. Figure 3C: Please include what the X-axis is plotting? As a supplemental figure, please include your initial gating strategy used for all flow cytometry data. Are there any CD68+CD11b+ CD45 high cells in these animals? If an effect of IL-37 is seen in filtration of peripheral inflammatory macrophages into the brain, this is a very strong data set to improve the impact of this figure.

All detailed information on plot axis and gating strategies was added to the revised manuscript. In addition, the FACS data were reanalyzed to verify the populations of infiltrating macrophages expressing CD68 in the brains of WT and IL-37tg mice after LPS challenge. Supplementary Figure 3 is included in the manuscript. The corresponding information can be found in lines 227-238.

9. Results, page 7: The authors make the statement that IL-37 signaling has a significant impact on the pro-inflammatory activation of microglia in the brain. However, this is based on just the measurement of IL-1b and microglial numbers. The authors should measure other markers of inflammation that are known to go up in the brain upon LPS challenge. Maybe using an Elisa based multiplex assay such as MSD or Luminex to measure brain inflammatory cytokines will add to this data set.

In this study, we specifically analyzed IL-1ꞵ because the production of this cytokine depends on the activation of the NLRP3 inflammasome, which also includes IL-18 and the closely related IL-37; all of these cytokines belong to the IL-1 family. This cytokine is an important mediator of the inflammatory response, particularly upon LPS stimulation (Lopez-Castejon and Brough, 2011). Therefore, we focused on this selected known cytokine.

However, in the new experiment, we also measured the levels of other cytokines, including IL-6 and TNF-α, using Elisa and added the data to the revised manuscript in supplementary Figure 2.

For microglia, we also analyzed CD68 expression, which in combination with proinflammatory cytokines (IL-1ꞵ) in brain and cultures represent features of high activation and neuroinflammatory processes.

These points have been highlighted in the revised manuscript. The information can be found in lines 227-230.

10. Results, page 7: Figure 4A-4B: Was the LPS injection used for these experiments the same as outlined for figure 3? Please clarify this in the method section, especially since 2 different LPS treatment paradigms have been used for 4A-4B and 4G-4H.

Because Figure 4A-C are the electrophysiological experiments after LPS challenge in vivo, the timing of the experiment is the same as in Figure 3, but we have made this clearer in the revised manuscript. The methods and information for Figure 4G-I have been included in the Methods and Results sections. The information can be found in lines 291308 and 909-914.

11. The paradigm for the Y-maze testing (pre-treatment with IL-37, followed by IL1b), though in-line with the current study, is not biologically relevant. Since IL-37 will be upregulated as part of the neuroinflammatory response, have the authors tried to treat animals with IL-37 after IL-1b or concurrently with IL-1B?

Although we agree with the comment that the best therapeutic regimen for the anti-inflammatory agent is to use it after the onset of symptoms, in this part of the study we focused on recommending rIL-37 as a preventive strategy rather than as a therapeutic agent for acute inflammation. In addition, pro-inflammatory members of the IL-1 family were hypothesized to play an important role in the pathophysiology of sepsis, as serum levels of these cytokines were greatly increased in patients with sepsis. Moreover, injection of IL-1β elicits an acute and rapid response in the body (similar to septic shock), making the search for preventive strategies for these conditions of great importance compared with therapeutic strategies, because the immune response is very rapid and not gradual. We have tried to make this point clearer in the revised manuscript.

12. Figure 1B: This figure suggests that 1 i.p injection with rIL-1B completely abolishes any working memory In WT mice. This is surprising. Please clarify.

Acute systemic inflammation has been shown to cause deficits in working memory early in the disease course in dementia models. Moreover, systemic inflammation primarily impairs attention and cognitive flexibility, including working memory, rather than associative learning. Therefore, short-term memory formation may be more vulnerable to even mild inflammatory responses, as it has been shown that a low-dose injection of LPS, which elicits a very mild immune response and small changes (< 1 °C) in core body temperature, severely impairs working memory (Murray et al., 2012). This is not the case for long-term memory formation. We therefore hypothesized that a systemic injection of IL-1β, which induces strong systemic inflammation, can completely abolish working memory, which is also confirmed by our results. These points have been carefully highlighted in the Discussion section. The information can be found in lines 590-597.

13. For Figures6 and 7: Were both sexes used for APPPS1 studies? If so, please clarify in the methods section as to the number of males and females, given that female APPPS1 mice show more aggressive disease as compared to males.

Mainly male mice were used in this study. All detailed information about the sexes of the mice was clarified in the revised manuscript. The information can be found in lines 885886.

14. Figure 6. Please show all gating strategies as well as relevant gates drawn to identify CD11b+ CD45low microglia, as well as the methoxy X-04+ vs methoxy X-04- microglia. Is there a CD11b+ CD45high microglial subset at this age, since at earlier timepoints, these distinct microglial subsets are evident and have different activation profiles.

All detailed information on gating strategies is included in the revised manuscript (lines 382-393). Cx3cr1GFP mice were used in this particular experiment. Therefore, the gating strategies were performed for the positive FITC cell population. The detailed information on this and supplementary Figure 6 are included in the revised manuscript.

15. Please include details of how brain tissue was processed to prepare single-cell suspension for all FACS data. i.e: percoll based density gradients or myelin removal beads etc?

All detailed information on the preparation of single cell suspensions for FACS data has been accurately added to the revised manuscript. The information can be found in lines 994-997.

16. Authors discuss their methoxy-X04 data as indicative of "increased clearance" of plaques. Careful interpretation of this data set is required as this assay only measures Abeta "uptake". Since no measures of clearance have been assessed, please reword and/ reinterpret this in the results and Discussion section.

We fully agree with this comment. The explanation and interpretation of this experiment was adjusted accordingly in the revised manuscript. The information can be found in lines 382-393 and 682-684.

17. Please include representative images for plaque loads in the cortex that correlate with quantification in 6I. Also, please include a representative image for differences in plaque size.

All the representative images mentioned for specific brain regions and different plaque sizes were added to the revised manuscript. The information can be found in Supplementary Figure 5.

18. Figure 7A: Please mention the average age of the animals used in the 3 animal groups for MW testing. 9-12 months is a big age range for these animals and results will be skewed if one cohort is largely composed of older animals vs. younger.

Carefully age-matched mice were used in all experimental groups. Detailed information on the average age of the animals was included in the revised manuscript. The information can be found in Supplementary Figure 8.

19. The Discussion section is rather long and largely used to re-summarize results. The authors should reconsider using this section to put their data into context for the big-picture field of neurodegeneration. Some suggestions – How do the authors picture IL-37 signaling to play a role in human AD?a. Given the detailed description of distinct microglial subsets associated with amyloid vs. tau pathology in AD, do the authors anticipate IL-37 to have distinct effects depending on pathological outcomes?b. Literature has demonstrated that IL-1b, IL-6 and TNFa from reactive microglia may play a role in exacerbating tau-spread. The authors should discuss the implications of their findings in this lightc. Is anything is known about IL-37 signaling in other neuroinflammatory models (injury, MS etc)? A possible discussion of these in the context of neuroprotection may strengthen this section.

All of the above valuable points were included in the Discussion section of the revised manuscript.

20. in vitro phenotypes in Figure 1 were not closely connected with the following in vivo neuroinflammation phenotypes. Although three different neuroinflammation models were used, the detailed mechanisms for how IL-37 reduced neuroinflammation were not well addressed.

Although this study suggests the positive role of IL-37 in modulating microglial activations as a possible key mechanism in controlling neuroinflammation, the detailed mechanisms are not addressed in this manuscript. However, we believe that our findings obtained in various inflammatory models clearly demonstrate the beneficial role of IL-37 and thus may pave the way for future studies to discover the detailed mechanisms and possible neuroprotective potential of IL-37, especially with regard to therapy in humans. The detailed possible underlying mechanisms of the neuroinflammatory potential of IL37 were proposed in the Discussion section of the revised manuscript based on the recent literature examining the beneficial role of IL-37 in autism, MS and spinal cord injury.

In addition, we performed a single-cell RNA sequencing experiment in the microglial cell population of both WT and IL-37tg mice after LPS challenge. We selected several differentially expressed candidate genes that may be important as key molecules to decipher the IL-37 underlying mechanisms in the future. Figure 8 and supplementary Figure 7 are included in the revised manuscript.

21) Figure 3 and Figure 5 showed reduced pro-inflammatory cytokine production in brain lysates. However, glial cells other than microglia, such as astrocytes and infiltrating leukocytes could be sources of inflammatory cytokines. There was no evidence supporting that reduced cytokine levels were intrinsic to microglia as modelled in the in vitro system. Thus, in vitro and in vivo phenotypes appeared somewhat disconnected.

As noted by the reviewer, the role of other cells in the brain, including astrocytes, brain infiltrating macrophages, and lymphocytes, is not evident in the cytokine assay of brain lysate. However, reanalysis of our FACS data with gating strategies for infiltrating macrophages and lymphocytes revealed no differences between groups, suggesting a smaller population and no significant activation profile compared with microglia. Moreover, based on the new experiment performed with primary astrocyte cultures (see answer to question 1), the key role of microglia in this scenario is suggested. The relevant information was added to the manuscript (lines 171-175) and can be found in supplementary Figure 1 and 3.

22) In Figure 5, IL-1β-induced neuroinflammation model was used to demonstrate the beneficial effects of IL-37 on cognition and synaptic function. Could injection of recombinant IL-37 display similar effects for LPS challenge model in Figure 3 and Figure 4?

As mentioned by the editor, confirmation of the potential anti-inflammatory role of IL-37 in our in vivo model with LPS challenge is of great importance. We performed new experiments and pre-treated the mice with rIL-37 followed by LPS challenge. Subsequently, the level of pro-inflammatory cytokines in the brain was measured using Elisa. The new data were added to the revised manuscript. The information can be found in the manuscript (lines 335-346) and in supplementary Figure 4.

23) Reduced IL-37-mediated neuroinflammation was shown in both chronic and acute models in the manuscript. Did microglia or other cells play a major role in those models? The detailed mechanisms regarding cell type contribution were largely missing in the manuscript.

The detailed information on the importance of microglial function in neuroinflammatory processes and in the progression of Alzheimer's disease, based on the most recent findings, was added to the revised manuscript.

In addition, the new experiments on the role of astrocytes and other infiltrating immune cells in this scenario were added to the revised manuscript. Please see the answers to questions 1 and 21.

24. Figure 2 showed differential metabolomic profiling of microglial cells between WT and IL-37tg mice. However, there was no further evidence demonstrating that the metabolic function of microglial cells was indeed altered by IL-37 expression. It would be better to show the results of seahorse or other metabolic functional assays.

In this study, we focused on the two metabolites that have been shown to be associated with inflammation in macrophages. We fully agree with the reviewers that further experiments showing metabolic alterations could be useful to address the subsequent changes in more detail. However, it has already been pointed out that itaconate and succinate in particular are involved in inflammation, especially in macrophages. Further data on the expression level of the *Acod1* gene specifically in microglia, which is responsible for itaconate production, was also added to the revised manuscript. There was no significant differences in *Acod1* expression after LPS challenge between WT and IL-37tg mice, which may be due to the degradation of this transcript that has already occurred (although *Acod1* was still slightly less expressed in the IL-37tg mice). The information can be found in the manuscript (lines 622-636) and in supplementary Figure 7.

25. The models for acute neuroinflammation including LPS and IL-1β challenge were systematic inflammation. It might be reasonable to propose that reduced neuroinflammation was a secondary effect to reduced inflammation response in the periphery. In addition, in Figure 3, would injection of LPS twice induce tolerance responses?

We agree with the reviewer that containment of the inflammatory response in the periphery by IL-37 could prevent the inflammatory consequences in the brain (please see the answer to question 3).

As mentioned in the comment, the immune tolerance phenomenon is very important in this study. The protocol for 2-times LPS injection was adopted from the study by Wendeln et al. 2018. Their results clearly showed that tolerance in the peripheral immune response occurred after the 2-times LPS injection, as indicated by the cytokine expression levels in the serum. Remarkably, however, this was not the case for the CNS, as cytokine levels in the brain were significantly increased after the second LPS injection.

26. In Figure 3 and Figure 6, CD68 was used as an activation marker for microglia. However, CD68 expression by itself is not enough to define microglia to be in the activation state. The phenotypic changes of microglial cells would also depend on specific models used. Additional experimental evidence is needed for defining the reduced activation status of microglia in IL-37tg mice.

CD68 is a lysosomal protein that is highly expressed by activated macrophages and activated microglia and expressed to a very low extent by resting microglia, as evidenced by much of the literature (Urga et al. 2020, Kettenmann et al. 2011 or Yang et al. 2013). Therefore, CD68 can be considered a well-established activation marker for microglia. However, in this study, our results additionally showed a lower number of IBA-1-positive cells and lower IL-1β expression in the brain of IL-37tg mice after LPS challenge, which may confirm the reduced activation status of microglia in IL-37tg mice. The relevant literature on CD68 as an activation marker for microglia were added to the revised manuscript.